# SolveSAPHE-r2 (v2.0.1): revisiting and extending the Solver Suite for Alkalinity-PH Equations for usage with $CO_2$, $HCO_3^-$ or $CO_3^{2-}$ input data

Guy Munhoven

Dépt. d'Astrophysique, Géophysique et Océanographie, Université de Liège, B–4000 Liège, Belgium

**Correspondence:** Guy Munhoven
(Guy.Munhoven@uliege.be)

**Abstract.** The successful and efficient approach at the basis of SolveSAPHE (Munhoven, 2013), which determines the carbonate system speciation by calculating $p$H from total alkalinity ($\mathrm{Alk_T}$) and dissolved inorganic carbon ($C_\mathrm{T}$), and which converges for any physically sensible pair of such data, has been adapted and further developed to work with $\mathrm{Alk_T}$ & $CO_2$, $\mathrm{Alk_T}$ & $HCO_3^-$ and $\mathrm{Alk_T}$ & $CO_3^{2-}$. The mathematical properties of the three modified alkalinity-$p$H equations are explored. It is shown that
the $\mathrm{Alk_T}$ & $CO_2$ and $\mathrm{Alk_T}$ & $HCO_3^-$ problems have one and only one positive root for any physically sensible pair of data (i.e, such that $[CO_2] > 0$ and $[HCO_3^-] > 0$). The space of $\mathrm{Alk_T}$ & $CO_3^{2-}$ pairs is partitioned into regions where there is either no solution, one solution or where there are two. The numerical solution of the modified alkalinity-$p$H equations is far more demanding than that for the original $\mathrm{Alk_T}$-$C_\mathrm{T}$ pair as they exhibit strong gradients and are not always monotonous. The two main algorithms used in SolveSAPHE v1 have been revised in depth to reliably process the three additional data input pairs.
The $\mathrm{Alk_T}$ & $CO_2$ pair is numerically the most challenging. With the Newton-Raphson based solver, it takes about five times as long to solve as the companion $\mathrm{Alk_T}$ & $C_\mathrm{T}$ pair; the $\mathrm{Alk_T}$ & $CO_3^{2-}$ pair requires on average about four times as much time as the $\mathrm{Alk_T}$ & $C_\mathrm{T}$ pair. All in all, the secant based solver offers the best performances. It outperforms the Newton-Raphson based one by up to a factor of four in terms of average numbers of iterations and execution time and yet reaches equation residuals that are up to seven orders of magnitude lower. Just like the $p$H solvers from the v1 series, SolveSAPHE-r2 includes automatic
root bracketing and efficient initialisation schemes for the iterative solvers. For $\mathrm{Alk_T}$ & $CO_3^{2-}$ data pairs, it also determines the number of roots and calculates non-overlapping bracketing intervals. An open source reference implementation of the new algorithms in Fortran 90 is made publicly available for usage under the GNU Lesser General Public Licence version 3 (LGPLv3) or later.

## 1   Introduction

Among all the aspects of the ongoing global environmental changes (climate change, ocean acidification, . . . ), the solution chemistry of carbon dioxide ($CO_2$) is one of the best known. The related chemistry of the carbonate system in the oceans and other aqueous environments is well understood and routinely monitored and modelled. The equilibria between the carbonate system species involves four variables: $[CO_2]$ (or equivalently the partial pressure of $CO_2$, $p\mathrm{CO_2}$, or its fugacity,

$f\text{CO}_2$), $[\text{HCO}_3^-]$, $[\text{CO}_3^{2-}]$ and $[\text{H}^+]$ (or equivalently $p\text{H}$). The speciation, i.e., the determination of the concentrations of the individual species, therefore also requires four constraints. Two of these are given by the equilibrium relationships that characterize the equilibria between dissolved $\text{CO}_2$ and $\text{HCO}_3^-$ on one hand, and between $\text{HCO}_3^-$ and $\text{CO}_3^{2-}$ on the other hand, assuming that the respective equilibrium constants are known or can be calculated. Two more independent constraints are thus required to completely characterize the system. These two additional constraints are generally chosen among the four traditional measurables of the system (see, e.g., Dickson et al., 2007): (1) the total concentration of dissolved inorganic carbon, $C_\text{T} = [\text{CO}_2] + [\text{HCO}_3^-] + [\text{CO}_3^{2-}]$; (2) total alkalinity, $\text{Alk}_\text{T}$; (3) $p\text{H}$ and (4) $p\text{CO}_2$ or $f\text{CO}_2$. Recently, a procedure to measure $[\text{CO}_3^{2-}]$ has been developed as well, thus increasing the number of measurables to five (Byrne and Yao, 2008; Patsavas et al., 2015; Sharp and Byrne, 2019). This latter has, however, not yet been widely adopted. With these two additional constraints, the concentrations of all the individual species as well as $C_\text{T}$ & $\text{Alk}_\text{T}$ can then be calculated.

There are ten different data pairs that can be composed from the set of five independent measurable variables of the carbonate system; there are fifteen if we further include $\text{HCO}_3^-$ as a sixth independent variable, although currently not (yet) measurable. Most modellers will call upon $C_\text{T}$ and $\text{Alk}_\text{T}$ which, besides being measurable, are also conservative and thus convenient for a budgeting approach. Experimentalists will use the pair that best suits their analytical equipment and expertise. Depending on the study requirements, not all pairs are equally attractive though. The analysis of Sharp and Byrne (2019) reveals that it is always advisable to measure $p\text{CO}_2$ directly if that variable is required: the uncertainty of calculated $p\text{CO}_2$ is always five to ten times as large as that of the directly measured one. $[\text{CO}_3^{2-}]$ can be calculated with lower uncertainty from $\text{Alk}_\text{T}$ & $C_\text{T}$ data than it can be directly measured; calculating it from other data pairs always bears greater uncertainty than directly measuring it. The most peculiar combination of uncertainties affects the results derived from paired measurements of $p\text{CO}_2$ and $\text{CO}_3^{2-}$: they allow to calculate $p\text{H}$ with the same uncertainty as if directly measured, thus providing nearly optimal values for the three individually measurable species. The uncertainties of the $\text{Alk}_\text{T}$ and $C_\text{T}$ calculated from it are, however, about twenty times as large as if directly measured. The currently most attractive pairs are $\text{Alk}_\text{T}$ & $p\text{CO}_2$ and $C_\text{T}$ & $p\text{CO}_2$, both of which allow to calculate $p\text{H}$ with better and $[\text{CO}_3^{2-}]$ with only slightly larger but still acceptable uncertainty than the direct measurement would offer. Direct $[\text{CO}_3^{2-}]$ measurements, which might be most advisable for tracing carbonate mineral saturation states, are best paired with $\text{Alk}_\text{T}$ or $C_\text{T}$ (Sharp and Byrne, 2019). It can nevertheless be expected that, once it becomes more widely used, the measurement uncertainty currently affecting that still young measurable can be reduced and eventually become better than that of $[\text{CO}_3^{2-}]$ calculated from $\text{Alk}_\text{T}$ & $C_\text{T}$, which is currently the best option (Sharp and Byrne, 2019).

Eleven out of these fifteen possible pairs of independent parameters of the carbonate system can be directly solved or require at most the resolution of a quadratic equation. The remaining four pairs require iterative procedures. Besides the $\text{Alk}_\text{T}$ & $C_\text{T}$ pair which was addressed in full detail by Munhoven (2013) these are (1) $\text{Alk}_\text{T}$ & $\text{CO}_2$, (2) $\text{Alk}_\text{T}$ & $\text{HCO}_3^-$ and (3) $\text{Alk}_\text{T}$ & $\text{CO}_3^{2-}$. Such calculations are performed to an advanced level of detail with dedicated and highly specialised packages. The review of Orr et al. (2015) offers a systematic analysis of subsisting uncertainties and inconsistencies between ten such packages, focusing on the sets of equilibrium constants adopted, pressure corrections applied, etc. Here, we do not focus on these aspects, but on the design of algorithms that can solve the underlying mathematical problem with as little user input as possible. The aim is to reduce user input to the bare essentials: besides the fundamental information about temperature, salinity,

pressure and the thermodynamic data, this ideally had to be any physically meaningful data pair only; the algorithm should be
able to derive any other auxiliary information, such as root brackets or starting values for iterations, on its own.

In the companion paper (Munhoven, 2013), such autonomous algorithms with robust convergence properties for a wide range of environmental conditions are presented for usage with the $Alk_T$ & $C_T$ pair. Here, we are revisiting that approach, extending and adapting it so that the $Alk_T$ & $CO_2$, $Alk_T$ & $HCO_3^-$ and $Alk_T$ & $CO_3^{2-}$ pairs can be processed with the same ease and reliability. For the sake of completeness – and with minimal details only – "recipes" for solving the other eleven explicit cases
are provided in the appendix. Alternative approaches can be found in the literature, such as in Zeebe and Wolf-Gladrow (2001) or Dickson et al. (2007). Dickson et al. (2007) also provide pathways for using triplets or quartets of input data, which only require the knowledge of one of the two dissociation constants or of their ratio, or none of them. That kind of approach is, however, not considered in this study.

## 2   Theoretical Considerations

In the following, it is assumed that the temperature $T$, salinity $S$ and applied pressure $P$ are given and that adequate values for all the required stoichiometric equilibrium constants are available. It is furthermore assumed that the total concentrations of all the other relevant acid systems (borate, hydrogen sulfate, phosphate, silicate, etc.) are known.

### 2.1   Revisiting the mathematics of the alkalinity-*p*H equation

Cornerstone to the speciation calculation is the resolution of the following equation, which I call the *alkalinity-pH equation* as
it derives from the definition of total alkalinity:

$$R_T([H^+]) \equiv Alk_{nW}([H^+]) + \frac{K_W}{[H^+]} - \frac{[H^+]}{s} - Alk_T = 0, \tag{1}$$

i.e., Eq. (21) from Munhoven (2013). In this equation, $[H^+]$ is the proton concentration expressed on one of the commonly used *p*H scales (total, seawater) and $s$ is a factor to convert from that scale to the free scale. $s$ depends on temperature, pressure and salinity of the sample and its value is close to 1 (typically between 1.0 and 1.3). The first term at the right-hand side is that part of
the total alkalinity that is not related to the water self-ionization: $Alk_{nW}([H^+]) = \sum_i Alk_{A_{[i]}}([H^+])$, where $i$ denumbers the acid systems resulting from the dissolution of acids $A_{[i]}$ whose dissolution products contribute to total alkalinity. For the purpose of this study, $Alk_{nW}([H^+])$ is partitioned into carbonate alkalinity, $Alk_C([H^+])$, and non-carbonate alkalinity, $Alk_{nWC}([H^+])$,

$$Alk_{nW}([H^+]) = Alk_C([H^+]) + Alk_{nWC}([H^+])$$

since the relevant carbonate system parameters (the concentrations of $CO_2$, $HCO_3^-$ and $CO_3^{2-}$ and their sum, $C_T$) are all
directly related to $Alk_C$. Similarly to $Alk_{nW}$, $Alk_{nWC}$ admits an infimum and a supremum which can both be derived from the total concentrations of all the acid-base systems considered. We denote these two by $Alk_{nWCinf}$ and $Alk_{nWCsup}$, respectively. Eq. (1) is thus formally rewritten as

$$R_T([H^+]) \equiv Alk_C([H^+]) + Alk_{nWC}([H^+]) + \frac{K_W}{[H^+]} - \frac{[H^+]}{s} - Alk_T = 0, \tag{2}$$

The carbonate alkalinity term writes, as a function of $C_T$

$$\text{Alk}_C([H^+]) = \frac{K_1[H^+] + 2K_1K_2}{[H^+]^2 + K_1[H^+] + K_1K_2} C_T \tag{3}$$

where $K_1$ and $K_2$ are the first and second stoichiometric dissociation constants of carbonic acid. The individual carbonate species fractions of $C_T$ can be expressed as a function of $[H^+]$:

$$[CO_2] = \frac{[H^+]^2}{[H^+]^2 + K_1[H^+] + K_1K_2} C_T \tag{4}$$

$$[HCO_3^-] = \frac{K_1[H^+]}{[H^+]^2 + K_1[H^+] + K_1K_2} C_T \tag{5}$$

$$[CO_3^{2-}] = \frac{K_1K_2}{[H^+]^2 + K_1[H^+] + K_1K_2} C_T. \tag{6}$$

Accordingly, $\text{Alk}_C([H^+])$ may be rewritten in one of the following forms

$$\text{Alk}_C([H^+]) = \frac{K_1[H^+] + 2K_1K_2}{[H^+]^2} [CO_2] \tag{7}$$

$$\text{Alk}_C([H^+]) = \frac{K_1[H^+] + 2K_1K_2}{K_1[H^+]} [HCO_3^-] \tag{8}$$

$$\text{Alk}_C([H^+]) = \frac{K_1[H^+] + 2K_1K_2}{K_1K_2} [CO_3^{2-}]. \tag{9}$$

In order to get a first idea about the complications that we might encounter for the solution of the three new data pairs, we start with an exploratory analysis using SolveSAPHE version 1.0.3 (Munhoven, 2013–2021). The three panels in the upper row of Fig. 1 show the $pH$ and the $HCO_3^-$ and $CO_3^{2-}$ concentration distributions for a reduced SW3 test case from Munhoven (2013), with the $C_T$ range extending from 0 to $4\,\text{mmol kg}^{-1}$ and the $\text{Alk}_T$ range from $-1$ to $3\,\text{mmol kg}^{-1}$ only. The $pH$ distribution from Fig. 1a is then used to derive the corresponding $CO_2$ (not shown), $HCO_3^-$ and $CO_3^{2-}$ concentration distributions (Figs. 1b and c). Since we intend to solve the alkalinity-$pH$ equation for given $\text{Alk}_T$, and either one of $[CO_2]$, $[HCO_3^-]$ or $[CO_3^{2-}]$, we furthermore produce the concentration isolines for the three species on a $pH$-$\text{Alk}_T$ graph (Figs. 1d, e and f). For these three latter, we first calculated $\text{Alk}_C$ from $\text{Alk}_T$ by using Eq. (2). Positive $\text{Alk}_C$ were then used with Eqs. (7), (8) and (9) to derive the corresponding $[CO_2]$, $[HCO_3^-]$ and $[CO_3^{2-}]$, respectively. Blank areas represent the $pH$-$\text{Alk}_T$ combinations that lead to negative $\text{Alk}_C$.

The V- or U-shaped isolines for $HCO_3^-$ on the $C_T$-$\text{Alk}_T$ graph and for $CO_3^{2-}$ on the $C_T$-$\text{Alk}_T$ and on the $pH$-$\text{Alk}_T$ graphs show that the $C_T$ & $HCO_3^-$ and the $\text{Alk}_T$ & $CO_3^{2-}$ pairs will not always provide unambiguous results. This is illustrated by the blue and the black stars in Figs. 1c and 1f: they both lie on the $100\,\mu\text{mol kg}^{-1}$ $[CO_3^{2-}]$ isoline and on the horizontal line drawn through $\text{Alk}_T = 2.3\,\text{mmol kg}^{-1}$. For that pair of data values there are thus two compatible $C_T$ and correspondingly two possible $pH$ values. On the other hand, with the same $\text{Alk}_T$, a $[CO_3^{2-}]$ of $1\,\text{mmol kg}^{-1}$ would not provide any solution as the $1\,\text{mmol kg}^{-1}$ isoline has its minimum at $\text{Alk}_T = 2.63\,\text{mmol kg}^{-1}$. Similarly, there are pairs of $C_T$ and $[HCO_3^-]$ values that are compatible with two $\text{Alk}_T$ values and thus two $pH$ values, others with none: a vertical line drawn through $C_T = 2.2\,\text{mmol kg}^{-1}$ crosses the $2.0\,\text{mmol kg}^{-1}$ isoline for $[HCO_3^-]$ twice and so that pair of data values leads to two $pH$ solutions; a vertical line

drawn through $C_T = 2.05\,\mathrm{mmol\,kg^{-1}}$ does not cross that $2.0\,\mathrm{mmol\,kg^{-1}}$ isoline for $[\mathrm{HCO_3^-}]$ at all and that pair of data values does not have any $p\mathrm{H}$ solution.

As will be shown below, the SolveSAPHE approach of Munhoven (2013), which is based upon the use of a hybrid iterative solver safeguarded by intrinsic brackets that can be calculated a priori, can be easily adapted for the $\mathrm{Alk_T}$ & $\mathrm{CO_2}$ and $\mathrm{Alk_T}$ & $\mathrm{HCO_3^-}$ pairs. According to the outcome of our preliminary analysis above, the $\mathrm{Alk_T}$ & $\mathrm{CO_3^{2-}}$ pair requires a more in-depth analysis. We show that it is nevertheless possible to diagnose the different cases that can theoretically be encountered and, in case there are two solutions, to derive bracketing intervals for each of the two and to isolate them efficiently. For each pair, we

(1) establish the analytical properties of the modified alkalinity-$p\mathrm{H}$ equation; (2) derive brackets for the root(s); (3) develop a reliable and safe algorithm to solve the problem; (4) design an efficient initialisation scheme. The $C_T$ & $\mathrm{HCO_3^-}$ pair, which requires only a quadratic equation to be solved, is straightforwoard to diagnose a priori (see the corresponding recipe in the appendix).

     We will now in turn analyse the mathematical properties of the alkalinity-$p\mathrm{H}$ equation that results from the substitution of

$C_T$ by the concentration of one of its individual species.

## 2.2   The $\mathrm{Alk_T}$ & $\mathrm{CO_2}$ Problem

### 2.2.1   Mathematical Analysis

The $\mathrm{Alk_T}$ & $\mathrm{CO_2}$ pair can be dealt with in a similar way to the $\mathrm{Alk_T}$ & $C_T$ pair in the original SolveSAPHE. The $\mathrm{Alk_C}([\mathrm{H^+}])$ term in Eq. (2) is written as in Eq. (7). Equation (2) then becomes

$$\left(\frac{K_1}{[\mathrm{H^+}]} + \frac{2K_1 K_2}{[\mathrm{H^+}]^2}\right)[\mathrm{CO_2}] + \mathrm{Alk_{nWC}}([\mathrm{H^+}]) + \frac{K_W}{[\mathrm{H^+}]} - \frac{[\mathrm{H^+}]}{s} - \mathrm{Alk_T} = 0. \tag{10}$$

Just like the $\mathrm{Alk_C}([\mathrm{H^+}])$ expression from Eq. (3) is monotonously decreasing with $[\mathrm{H^+}]$ for $C_T$ fixed, that from Eq. (7) is monotonously decreasing with $[\mathrm{H^+}]$ for $[\mathrm{CO_2}]$ fixed. The expression at the left-hand side of Eq. (10) decreases from $+\infty$ to $-\infty$ for $[\mathrm{CO_2}] > 0$ as $[\mathrm{H^+}]$ varies from $0^+$ to $+\infty$. Eq. (10) thus always has exactly one positive solution.

### 2.2.2   Root bracketing

Intrinsic brackets for the solution of Eq. (10) can be derived similarly to what is done in Sect. 5.1 in Munhoven (2013). The lower bound $H_{\mathrm{inf}}$ can be chosen such that

$$\left(\frac{K_1}{H_{\mathrm{inf}}} + \frac{2K_1 K_2}{H_{\mathrm{inf}}^2}\right)[\mathrm{CO_2}] + \frac{K_W}{H_{\mathrm{inf}}} - \frac{H_{\mathrm{inf}}}{s} = \mathrm{Alk_T} - \mathrm{Alk_{nWCinf}}$$

i.e., as the positive root of the cubic equation

$$\frac{H^3}{s} + (\mathrm{Alk_T} - \mathrm{Alk_{nWCinf}})H^2 - (K_1[\mathrm{CO_2}] + K_W)H - 2K_1 K_2[\mathrm{CO_2}] = 0$$

Let us denote this cubic by $P(H)$. It is important to notice that $P(0) = -2K_1 K_2[\mathrm{CO_2}] < 0$ and $P'(0) = -(K_1[\mathrm{CO_2}] + K_W) < 0$. The equation $P(H) = 0$ has therefore one and only one positive root.

Similarly, the upper bound $H_{\text{sup}}$ can be chosen such that

$$\left(\frac{K_1}{H_{\text{sup}}} + \frac{2K_1 K_2}{H_{\text{sup}}^2}\right)[\text{CO}_2] + \frac{K_{\text{W}}}{H_{\text{sup}}} - \frac{H_{\text{sup}}}{s} = \text{Alk}_{\text{T}} - \text{Alk}_{\text{nWCsup}}$$

i.e., as the positive root of the cubic equation

$$\frac{H^3}{s} + (\text{Alk}_{\text{T}} - \text{Alk}_{\text{nWCsup}})H^2 - (K_1[\text{CO}_2] + K_{\text{W}})H - 2K_1 K_2[\text{CO}_2] = 0$$

which has also one and only one positive root, for the same reasons as above.

The positive roots of these cubic equations can be found by adopting a strategy similar to that used for the cubic initialisation of the iterative solution in SolveSAPHE (Munhoven, 2013, Sect. 3.2.2):

1. Locate the local minimum of the cubic, in $H_{\text{min}} > 0$;

2. Develop the cubic as a quadratic Taylor expansion, $Q(H)$, around that minimum;

3. Solve $Q(H) = 0$ which has two roots and chose the one that is greater than $H_{\text{min}}$.

In this particular case, it is, however, not necessary to solve these equations exactly as we only need approximate bounds of the root for safeguarding the iterations while solving Eq. (2). For $H_{\text{inf}}$ we may actually chose the $H_{\text{min}}$ of the first cubic which is lower than the positive root and thus sufficient. Regarding $H_{\text{sup}}$, it should be noticed that $P(H) = Q(H) + (H - H_{\text{min}})^3/s$.
Accordingly, $P(H) > Q(H)$ for $H > H_{\text{min}}$ and therefore the greater root of $Q(H)$ for the second cubic is greater than the positive root of that cubic. The greater of the two roots of $Q(H)$ is therefore a sufficient upper bracket and may be used instead of the exact $H_{\text{sup}}$.

Any bracketing root-finding algorithm can then be used to solve the modified alkalinity $p$H equation (10).

## 2.3 The $\text{Alk}_{\text{T}}$ & $\text{HCO}_3^-$ Problem

### 2.3.1 Mathematical analysis

For the $\text{Alk}_{\text{T}}$ & $\text{HCO}_3^-$ pair, the $\text{Alk}_{\text{C}}([\text{H}^+])$ term in Eq. (2) is written as in Eq. (8):

$$\left(1 + \frac{2K_2}{[\text{H}^+]}\right)[\text{HCO}_3^-] + \text{Alk}_{\text{nWC}}([\text{H}^+]) + \frac{K_{\text{W}}}{[\text{H}^+]} - \frac{[\text{H}^+]}{s} - \text{Alk}_{\text{T}} = 0. \tag{11}$$

The expression at the left-hand side of Eq. (11) decreases monotonuously from $+\infty$ to $-\infty$ for $[\text{HCO}_3^-] > 0$ fixed as $[\text{H}^+]$ varies from $0^+$ to $+\infty$. Equation (11) thus always has exactly one positive solution.

### 2.3.2 Root bracketing

The lower bound $H_{\text{inf}}$ can be chosen such that

$$\left(1 + \frac{2K_2}{H_{\text{inf}}}\right)[\text{HCO}_3^-] + \frac{K_{\text{W}}}{H_{\text{inf}}} - \frac{H_{\text{inf}}}{s} = \text{Alk}_{\text{T}} - \text{Alk}_{\text{nWCinf}}$$

i.e., as the positive root of the quadratic equation

$$\frac{H^2}{s} + (\text{Alk}_\text{T} - \text{Alk}_\text{nWCinf} - [\text{HCO}_3^-])H - (2K_2[\text{HCO}_3^-] + K_\text{W}) = 0.$$

Similarly, the upper bound $H_\text{sup}$ can be chosen such that

$$\left(1 + \frac{2K_2}{H_\text{sup}}\right)[\text{HCO}_3^+] + \frac{K_\text{W}}{H_\text{sup}} - \frac{H_\text{sup}}{s} = \text{Alk}_\text{T} - \text{Alk}_\text{nWCsup}$$

i.e., as the positive root of the quadratic equation

$$\frac{H^2}{s} + (\text{Alk}_\text{T} - \text{Alk}_\text{nWCsup} - [\text{HCO}_3^-])H - (2K_2[\text{HCO}_3^-] + K_\text{W}) = 0.$$

Both equations always have two roots, one positive and one negative — their product is negative as indicated by the constant
term. With the respective positive roots, we have again bounds for the solution of the modified alkalinity-$p$H equation and any bracketing root-finding algorithm can be used to solve it.

## 2.4 The Alk$_\text{T}$ & CO$_3^{2-}$ Problem

Whereas any physically meaningful Alk$_\text{T}$-[CO$_2$] or Alk$_\text{T}$-[HCO$_3^-$] concentration pairs will always provide one and only one [H$^+$] (or equivalently $p$H) value as demonstrated above, this cannot be the case for every Alk$_\text{T}$-[CO$_3^-$] pair, as can be deduced
from Figs. 1c and 1f. On one hand, there are two compatible $C_\text{T}$, and equivalently two $p$H values, for most Alk$_\text{T}$-[CO$_3^{2-}$] pairs. This little-known fact was already documented in the 1960s (see, e.g., Deffeyes (1965)).[1] On the other hand, there are also Alk$_\text{T}$-[CO$_3^{2-}$] pairs that do not allow for any solution, as they lead to negative carbonate alkalinity. To our best knowledge, none of the currently available carbonate system speciation programs takes this possibility into account.

### 2.4.1 Mathematical analysis and root bracketing

The solution of the Alk$_\text{T}$ & CO$_3^{2-}$ problem thus requires a more in-depth mathematical analysis. To start, we write out Eq. (2) with the Alk$_\text{C}$ expression for [CO$_3^{2-}$] (Eq. (9)):

$$\frac{K_1[\text{H}^+] + 2K_1K_2}{K_1K_2}[\text{CO}_3^{2-}] + \text{Alk}_\text{nWC}([\text{H}^+]) + \frac{K_\text{W}}{[\text{H}^+]} - \frac{[\text{H}^+]}{s} - \text{Alk}_\text{T} = 0.$$

Let us collect all the terms that are related to carbonate or water self-ionization alkalinity at the left-hand side, introduce the shorthand

$$\gamma = \frac{[\text{CO}_3^{2-}]}{K_2} - \frac{1}{s}.$$

---

[1]Zeebe and Wolf-Gladrow (2001) appear to be aware of it. In their recipe for given Alk$_\text{T}$ and [CO$_3^{2-}$] (on pp. 276–277), they indicate that the quintic equation to solve with their practical alkalinity approximation has two positive and three negative roots and that the larger positive one should be used (without any further justification, though). As shown here, this statement is not universally true – there are instances where that equation has only one positive or no positive roots. It is nevertheless true for typical seawater and the lower of the two positive roots actually implies unrealistically low, yet physically sensible, $C_\text{T}$ (see discussion in Sect. 2.4.2 below).

and rewrite the equation as

$$\gamma[\mathrm{H}^+] + \frac{K_\mathrm{W}}{[\mathrm{H}^+]} + 2[\mathrm{CO}_3^{2-}] = \mathrm{Alk_T} - \mathrm{Alk_{nWC}}([\mathrm{H}^+]). \tag{12}$$

The value of $\gamma$ is one of the main controls on the number of roots that this equation has.

1. If $\gamma < 0$, the equation has similar mathematical characteristics as the usual alkalinity-$p$H equation (Eq. (1). It has exactly one root which can be calculated using similar procedures as in the original SolveSAPHE. Please notice though that this means that $[\mathrm{CO}_3^{2-}] < \frac{K_2}{s}$. Since $K_2$ is of the order of $10^{-9}\,\mathrm{mol(kg\text{-}SW)}^{-1}$ and $s$ is of the order of 1, this case is only relevant for $\mathrm{CO}_3^{2-}$ concentrations of the order of $1\,\mathrm{nmol(kg\text{-}SW)}^{-1}$ and less.

2. If $\gamma = 0$ (i.e., if $[\mathrm{CO}_3^{2-}] = \frac{K_2}{s}$), the equation has exactly one root if $\mathrm{Alk_T} - 2[\mathrm{CO}_3^{2-}] - \mathrm{Alk_{nWCinf}} > 0$, no root otherwise.

3. If $\gamma > 0$, the left-hand side is not monotonous: it decreases from $+\infty$ in $[\mathrm{H}^+] = 0^+$ to a minimum (see below) and then increases back to $+\infty$ as $[\mathrm{H}^+] \to +\infty$. The right-hand side is bounded and strictly increasing over the same interval (Munhoven, 2013). As a result, the equation has no root if the right-hand side is too low, exactly one if the two curves become tangent and two roots if the right-hand side is great enough.

To alleviate notation let us define the two parametric functions

$$L([\mathrm{H}^+]; \gamma) = \gamma[\mathrm{H}^+] + \frac{K_\mathrm{W}}{[\mathrm{H}^+]} + 2[\mathrm{CO}_3^{2-}] \tag{13}$$

$$R([\mathrm{H}^+]; A) = A - \mathrm{Alk_{nWC}}([\mathrm{H}^+]), \tag{14}$$

where $[\mathrm{H}^+]$ is the independent variable and $\gamma$ and $A$ (alkalinity) are parameters. With these two function definitions, Eq. (12) then rewrites $L([\mathrm{H}^+]; \gamma) = R([\mathrm{H}^+]; \mathrm{Alk_T})$. Schematic representations of the three $\gamma$ cases and of the $L$ and $R$ functions are shown in Fig. 2.

**Case $\gamma < 0$**

The first case can be handled similarly to the $\mathrm{Alk_T}$ & $\mathrm{CO}_2$ and $\mathrm{Alk_T}$ & $\mathrm{HCO}_3^-$ pairs. Equation (12) always has exactly one root with $\gamma < 0$ as the equation function is monotonous and strictly decreasing with $[\mathrm{H}^+]$. Upper and lower bounds for that root can be derived by solving the (quadratic) equations

$$\gamma H_{\mathrm{inf}} + \frac{K_\mathrm{W}}{H_{\mathrm{inf}}} + 2[\mathrm{CO}_3^{2-}] = \mathrm{Alk_T} - \mathrm{Alk_{nWCinf}} \tag{15}$$

for $H_{\mathrm{inf}}$ and

$$\gamma H_{\mathrm{sup}} + \frac{K_\mathrm{W}}{H_{\mathrm{sup}}} + 2[\mathrm{CO}_3^{2-}] = \mathrm{Alk_T} - \mathrm{Alk_{nWCsup}} \tag{16}$$

for $H_{\mathrm{sup}}$, and retaining the respective positive roots of each.

**Case $\gamma = 0$**

The second case might be considered to be only mathematically of importance as it only applies for one exact (and thus improbable) $CO_3^{2-}$ concentration value. For the sake of completeness, I nevertheless solve it.

As mentioned above, if $\gamma = 0$, Eq. (12) has one solution if and only if $\text{Alk}_T - \text{Alk}_{nWCinf} > 2[CO_3^{2-}]$, and no solution else. The root can be easily bracketed from below. It is sufficient to chose $H_{inf}$ such that

$$\frac{K_W}{H_{inf}} = \text{Alk}_T - 2[CO_3^{2-}] - \text{Alk}_{nWCinf}$$

leading to $L(H_{inf}; \gamma) - R(H_{inf}; \text{Alk}_T) > 0$. The analogue equation for $H_{sup}$, with $\text{Alk}_{nWCinf}$ replaced by $\text{Alk}_{nWCsup}$ (cf. eqs. (15) and (16)) does not work if $\text{Alk}_T - \text{Alk}_{nWCsup} \leq 2[CO_3^{2-}]$. The newly derived asymptotic approximation for $\text{Alk}_{nWC}([H^+])$ as $[H^+] \to +\infty$ (see the *Mathematical and Technical Details* report in the Supplement) nevertheless provides a means to derive an upper bound. It is sufficient to chose $H_{sup}$ such that

$$\frac{K_W}{H_{sup}} = \text{Alk}_T - 2[CO_3^{2-}] - \text{Alk}_{nWCinf} - \frac{\sum_i [\Sigma A_{[i]}] K_{1,[i]}}{H_{sup}}$$

where $i$ denumbers the acid systems considered, except for the carbonate system, $[\Sigma A_{[i]}]$ is the total amount of the acid $i$ dissolved and $K_{1,[i]}$ is the first dissociation constant of the acid system $i$. This equation always has a solution and, taking into account that

$$\text{Alk}_{nWC}([H^+]) < \text{Alk}_{nWCinf} + \frac{\sum_i [\Sigma A_{[i]}] K_{1,[i]}}{[H^+]},$$

which is valid for $[H^+] > 0$, it is straightforward to show that $L(H_{sup}; \gamma) - R(H_{sup}; \text{Alk}_T) < 0$ with this choice. Equation (12), which is equivalent to $L(H; \gamma) - R(H; \text{Alk}_T) = 0$ thus has one single root between $H_{inf}$ and $H_{sup}$.

**Case $\gamma > 0$**

The third case is the most commonly encountered, and the most challenging. With $\gamma > 0$, $L([H^+]; \gamma)$ has a minimum and the location of that minimum is a critical parameter in the analysis of this case. Let us denote the location of that minimum by $H_{min}$ and the value that $L$ takes there by $L_{min}$:

$$H_{min} = \sqrt{\frac{K_W}{\gamma}} \quad \text{and} \quad L_{min} = 2\sqrt{\gamma K_W} + 2[CO_3^{2-}].$$

There are two ranges of $\text{Alk}_T$ values where firm conclusions can be drawn right away.

1. If $R(H_{min}; \text{Alk}_T) > L_{min}$, i.e., if $\text{Alk}_T > L_{min} + \text{Alk}_{nWC}(H_{min})$, Equation (12) has two distinct roots, since $R(H; \text{Alk}_T)$ is bounded. Furthermore, the roots — let us provisionally denote the lower one $H_1$ and the greater one $H_2$ — are such that $H_1 < H_{min}$ and $H_2 > H_{min}$. $H_{min}$ can thus be used as an upper bracket for $H_1$ and as a lower bracket for $H_2$. However, if $\text{Alk}_T - \text{Alk}_{nWCsup} > L_{min}$, the abscissae of the intersection points $P_{LL}$ and $P_{LR}$ (see Fig. 2), which are solutions of

$$\gamma H + \frac{K_W}{H} = \text{Alk}_T - 2[CO_3^{2-}] - \text{Alk}_{nWCsup}$$

provide tighter brackets than $H_{min}$.

2. If $\text{Alk}_T - \text{Alk}_{nWCinf} \leq L_{min}$, i.e., if $\text{Alk}_T \leq L_{min} + \text{Alk}_{nWCinf}$, Eq. (12) does not have any roots.

For intermediate values of $\text{Alk}_T$, no firm quantitative statement regarding the root(s) of Eq. (12) can be made a priori. As $\text{Alk}_T$ decreases from $L_{min} + \text{Alk}_{nWC}(H_{min})$ to $L_{min} + \text{Alk}_{nWCinf}$, Eq. (12) will at first still have two roots, but both are greater than or equal to $H_{min}$. At some intermediate value, $L([H^+]; \gamma)$ and $R([H^+]; \text{Alk}_T)$ become tangent. At this point, Eq. (12) has one double root, which is the abscissa of that tangent point, $H_{tan}$. $H_{tan}$ is actually a universally valid separation limit between two roots, if there are any. For lower values of $\text{Alk}_T$, the problem does not have any solutions.

The limiting $\text{Alk}_T$ value for which the two curves are tangent and the corresponding $H_{tan}$ value can be calculated with a common algorithm to characterize a bracketed local minimum, such as Brent's algorithm (Brent, 1973). To start, we reconsider $L([H^+]; \gamma) - R([H^+]; A) = 0$ not as an equation in $[H^+]$ for given parameter values $\gamma$ (or, equivalently, $[CO_3^{2-}]$) and $A$, but rather as an implicit definition for $A$ as a function of $[H^+]$, for a given $\gamma$ (here $\gamma > 0$). This implicit function definition can actually be solved explicitly here:

$$A([H^+]) = L([H^+]; \gamma) + \text{Alk}_{ncW}([H^+]).$$

Figure 3 shows how the two problems are related and which information can be derived from the analysis of $L([H^+]; \gamma)$ and $R([H^+]; A)$ to contribute to the solution of the minimization of $A([H^+])$. The determination of $H_{tan}$ is costly, generally more costly than the subsequent resolution of the $p$H equation itself. As mentioned right at the beginning of this section, there are extended ranges of $\text{Alk}_T$ values for which the exact knowledge of $H_{tan}$ is not indispensable. In these situations $H_{min}$ may be a sub-optimal but nevertheless sufficient separation limit for the roots (or equal to the double root itself), and cheap to calculate. If available, $H_{tan}$ can be used as an upper bound for the lower and as a lower bound for the greater of the two roots. To start the minimization algorithm to derive $H_{tan}$, we can use the three characteristic $[H^+]$ values from Fig. 3 as initial conditions. These are $H_{min}$ together with the abscissae $H_L$ and $H_R$ of the intersection points between $L([H^+]; \gamma)$ and the horizontal line at $\text{Alk}_{min} - \text{Alk}_{nWCinf}$, which are the roots of

$$\gamma H^2 - (\text{Alk}_{min} - 2[CO_3^{2-}] - \text{Alk}_{nWCinf})H + K_W = 0.$$

By construction, $\text{Alk}_{min} - \text{Alk}_{nWCinf} > L_{min} = 2\sqrt{\gamma K_W} + 2[CO_3^{2-}]$. The discriminant of this quadratic equation is therefore strictly positive and the equation has two positive roots (their sum and their product are positive) as required. It is possible to show that the second derivative of $R([H^+]; A)$ with respect to $[H^+]$ is positive provided that the successive dissociation constants $K_{j,[i]}$ of the different acid systems (denumbered by $i$) resulting from the dissociation of an acid $H_{n_{[i]}}A_{[i]}$ are such that $K_{j,[i]} < \frac{1}{2}K_{j-1,[i]}, j = 2, \ldots, n_{[i]}$ — a very weak constraint as these constants generally generally differ by a few orders of magnitude. This has been verified to be the case for acid systems with $n_{[i]} = 1, \ldots, 12$ The underlying technical developments can be found in the *Mathematical and Technical Details* report in the Supplement. $R([H^+]; A)$ is thus concave, while $L([H^+]; \gamma)$ is convex for $\gamma > 0$. $A([H^+])$ thus has only one single local minimum comprised between $H_L$ and $H_R$.

Once $H_{tan}$ is known, the root brackets can be completed by the intersection points between $L([H^+]; \gamma)$ and the horizontal line at $\text{Alk}_T - \text{Alk}_{nWCinf}$ – corresponding to the $P_{UL}$ and $P_{UR}$ points in Fig. 2 with the grey band shifted down to include the minimum – i.e., by solving the same quadratic equation than for $H_L$ and $H_R$, with $\text{Alk}_{min}$ replaced by $\text{Alk}_T$. We have again

$\text{Alk}_\text{T} - \text{Alk}_\text{nWCinf} > L_\text{min}$ and the equation has two positive roots. With these brackets on the two roots, any safeguarded iterative procedure, such as those implemented in SolveSAPHE can be used to find the two roots in a controlled way.

### 2.4.2 Two roots: which one to chose?

Since every physical aqueous sample has a $pH$, the case without roots is essentially a theoretical one: it can actually arise only if the adopted alkalinity composition is not appropriate or if measurement errors are large. The case where an $\text{Alk}_\text{T}$ & $\text{CO}_3^{2-}$ data pair is compatible with two different $pH$ is, to the contrary, the most common one. SolveSAPHE-r2 has been designed as a universal $pH$ solver and as such returns all the roots that a data pair may offer, since there is no universal criterion to decide which one of the two roots is preferable over the other.

However, at the end of the calculations one of the two has to be chosen. Additional information, qualitative or quantitative is required to make that decision. This could be a third measurable, but often even qualitative information about, say, the expected $pH$ or the $C_\text{T}$ range might be sufficient. For typical seawater samples, the greater of the two $[\text{H}^+]$ solutions will typically be the adequate one, following the "use the larger one" advice of Zeebe and Wolf-Gladrow (2001)..

In the analysis of the $\text{Alk}_\text{T}$ & $\text{CO}_3^{2-}$ problem above, we determined the $\text{Alk}_\text{T}$ ranges that would respectively lead to two, one or no roots, for a given $\gamma$, i.e., $[\text{CO}_3^{2-}]$. To better understand the reasons why and when there are two, one or no roots and what other implications the individual roots have, it is instructive to perform the analysis the other way around: figure out how $[\text{CO}_3^{2-}]$ evolves as a function of $pH$, for a given value of $\text{Alk}_\text{T}$ and determine the $[\text{CO}_3^{2-}]$ ranges that would respectively lead to two, one or no roots. Such an analysis is presented in Fig. 4. For that figure, we have reconsidered the sample composition previously used in Figs. 1c and 1f. We thus start with $\text{Alk}_\text{T}$ fixed at $2.3\,\text{mmol}\,\text{kg}^{-1}$ and draw the evolution of $[\text{CO}_3^{2-}]$ as a function of $pH$ following

$$\text{CO}_3([\text{H}^+];\text{Alk}_\text{T}) = \frac{\text{Alk}_\text{T} - \text{Alk}_\text{nWC}([\text{H}^+]) - \frac{K_\text{W}}{[\text{H}^+]} + \frac{[\text{H}^+]}{s}}{\frac{[\text{H}^+]}{K_2} + 2}, \tag{17}$$

obtained by first using Eq. (9) to express $[\text{CO}_3^{2-}]$ as a function of $\text{Alk}_\text{C}$ and $[\text{H}^+]$, and then Eq. (2) to calculate $\text{Alk}_\text{C}$ as a function of $\text{Alk}_\text{T}$ and $[\text{H}^+]$ (and the total concentrations of all the other contributing acid-base systems, which we assume to be known, as stated initially). $\text{CO}_2$ and $\text{HCO}_3^-$ evolution curves can be derived similarly, by using Eqs. (7) and (8) resp. instead of (9). The concentration evolutions for the other $\text{Alk}_\text{T}$ contributors can be calculated from their species fraction equations (see, e.g., Munhoven, 2013). Figure 4a shows the concentration curves for all the species contributing to total alkalinity and dissolved inorganic carbon, for $pH$ ranging from 3 to 12, and for $\text{Alk}_\text{T} = 2.3\,\text{mmol}\,\text{kg}^{-1}$; Fig. 4b shows the $[\text{CO}_3^{2-}]$ evolution curves for different $\text{Alk}_\text{T}$ values, ranging from 0.5 to $2.5\,\text{mmol}\,\text{kg}^{-1}$. Solving the $\text{Alk}_\text{T}$ & $\text{CO}_3^{2-}$ problem for our showcase sample where $[\text{CO}_3^{2-}] = 0.1\,\text{mmol}\,\text{kg}^{-1}$ thus means drawing a horizontal line through the $0.1\,\text{mmol}\,\text{kg}^{-1}$ concentration level, and locating the intersection points with the $\text{CO}_3^{2-}$ curve, if any. There are actually two of them, located at $pH = 8.03$ and $pH = 11.43$. With increasing target values for $[\text{CO}_3^{2-}]$, i.e., moving the horizontal line upward, the two $pH$ roots will move closer and closer together, until the maximum of the $\text{CO}_3$ curve is touched, at $\text{CO}_{3\,\text{max}} = 841.7\,\mu\text{mol}\,\text{kg}^{-1}$. At this exact value, there will only be one root: $pH = 10.1943$. Positioning the line higher up does not allow any intersection with the $\text{CO}_3$ curve

any more: there are no roots for $[CO_3^{2-}] > CO_{3\,max}$. As illustrated in Fig. 4b, the value of $CO_{3\,max}$ grows as $Alk_T$ increases, thus extending the range of $[CO_3^{2-}]$ that allows for roots. Another noteworthy fact in Fig. 4b is that all the displayed $CO_3^{2-}$ curves increase to a maximum before declining and reducing to zero at some finite $pH$. While not all possible curves have that shape (it is possible to show that curves for $Alk_T < Alk_{nWCinf} + 2\frac{K_2}{s}$ are monotonously decreasing), all of them nevertheless go to zero.

The equation $CO_3([H^+]; Alk_T) = 0$ actually always has exactly one root, for any given $Alk_T$, since this simply requires that the numerator at the right-hand side of Eq. (17)) is 0, i.e., that $[H^+]$ is the solution of a standard alkalinity-$pH$ equation where $C_T = 0$. Such an equation always has exactly one positive solution, for any physically meaningful set of total concentrations of the different acid-base systems at play and any $Alk_T$ value (Munhoven, 2013). The $pH$ value at that zero-crossing point is furthermore the maximum possible $pH$ for the given $Alk_T$: beyond that value, the ever growing $[OH^+]$ would inevitably make

$Alk_T$ increase above the fixed value, thus requiring $Alk_C$ to become negative, which is not possible.

As can be seen in Fig. 4a, the high-$pH$ solution goes together with $C_T \simeq [CO_3^{2-}]$: $CO_3^{2-}$ represents only 4.6% of the $C_T$ at $pH = 8.03$, typical for seawater, but 99.2% at $pH = 11.43$. Accordingly $C_T = 0.1008\,\text{mmol kg}^{-1}$ at the high-$pH$ root, which is unrealistically low for many natural samples. In the marine realm, this observation regarding the high-$pH$ root is actually correct in general. $CO_3^{2-}$ represents more than 80% of DIC for $pH > 10$, and more than 90% for $pH > 10.3$, as can be calculated from

Eq. (6). In Fig. 4b, one can see that the maxima of the $CO_3$ curves are greater than $0.47\,\text{mmol kg}^{-1}$ for $Alk_T \geq 1.5\,\text{mmol kg}^{-1}$ and that they are located at $pH > 10$. Since the larger of the two solutions is always at greater or equal $pH$ than the maximum of the curve that it must intersect, we may conclude that for $[CO_3^{2-}] < 0.47\,\text{mmol kg}^{-1}$ and $Alk_T \geq 1.5\,\text{mmol kg}^{-1}$, the greater of the two $pH$ roots always implies that $CO_3^{2-}$ represents more than 80% of $C_T$. Accordingly, even a rough estimate of one of the other relevant parameters of the carbonate system might be sufficient to reject one of the two roots.

## 2.5   Initialisation: rationale

Since we have bracketing intervals for each diagnosed root, we may always use the fall-back initial value $H_0 = \sqrt{H_{inf} H_{sup}}$. This value is, however, often far from optimal. The efficient initialisation strategy of Munhoven (2013) can be generalized and adapted to each of the three pairs. For each case, we chose the most complex $Alk_T$ approximation that leads to a cubic equation. If the cubic polynomial behind that equation does not have a local minimum and a local maximum, we use the fall-back value.

If such a local minimum and maximum exist, we use the quadratic Taylor expansion around the relevant extremum — this will normally be the maximum if the coefficient of the cubic term is negative, and the minimum if that coefficient is positive. If that quadratic does not have any positive roots, the fall-back initial value is used. The roots for that quadratic are then determined. For problems that have only one positive $[H^+]$ solution ($Alk_T$ & $CO_2$, $Alk_T$ & $HCO_3^-$ and $Alk_T$ & $CO_3^{2-}$ with $\gamma < 0$), we consider that root of the quadratic expansion that is greater than the greatest location of the two extrema: if that root is lower

than $H_{inf}$, we use $H_0 = H_{inf}$; if it is greater than $H_{sup}$, we set $H_0 = H_{inf}$. For problems that have two positive $[H^+]$ solutions ($Alk_T$ & $CO_3^{2-}$ with $\gamma > 0$ and sufficiently great $Alk_T$), the initial value for determining the greater of the two $[H^+]$ solutions can be chosen exactly the same way; the initial value required to calculate the lower of the two $[H^+]$ solutions may be more tricky. If the location of the right-hand side extremum is too close to 0, the estimated root of the cubic may be negative. In this

case, the quadratic fitted to left-hand extremum should be considered as well and the greater of its roots tested. Because of the symmetries of a cubic, that root can be calculated with a few extra additions only.

The developments for each of the three input pairs are presented in full detail in the *Mathematical and Technical Details* report in the Supplement.

## 3 Numerical Experiments

### 3.1 Reference Fortran 90 implementation

The SolveSAPHE Fortran 90 library from Munhoven (2013) – hereafter SolveSAPHE v1 – has been revised, cleaned up and upgraded to allow the processing of the additional three pairs. For the purpose of this paper, only the two main solvers have been kept: these are `solve_at_general`, which uses a Newton-Raphson method, and `solve_at_general_sec`, which uses the secant method. Both can be still be used with the same Application Programming Interface (API) as in v1. The instances in SolveSAPHE-r2 are, however, only wrappers to the newly added Newton-Raphson based `solve_at_general2` and secant (or more precisely regula falsi) based `solve_at_general2_sec` both of which are able to process problems that have two roots. They return the number of roots of the problem, as well as their actual values, if any.

In the course of the developments related to the $Alk_T$ & $CO_2$ pair the Newton-Raphson based algorithm showed a few weaknesses. With the $Alk_T$ & $C_T$ pair that SolveSAPHE v1 had been designed for, each non-water alkalinity term was bounded, just like its derivative. Once $CO_2$ takes the role of $C_T$ these favourable properties are lost: with $[CO_2]$ fixed, the carbonate alkalinity term and its derivative with respect to $[H^+]$ become unbounded. Newton iterates can then change by large amounts and floating point over- and underflow errors on the exponential correction became common. The rate of change for Newton-Raphson iterates during each step was therefore limited to a factor of 100. With high $CO_2$ concentration values prescribed, there was another loss of control on the iteration sequence that had not been encountered before. At some iterations, most often at the first one, it happened that one of the two root brackets, say the upper one, was reduced to the iteration value. In the next iteration, that same bound was exceeded by the trial Newton-Raphson iterate, which was then rejected and replaced by a bisection iterate on the interval delimited by the previous iterate and the upper bracket. Since both were identical, the bisection actually produced no variation and falsely led to convergence diagnosis. This has been fixed by changing the interval whereon the bisection step is performed to that delimited by the lower and the upper brackets of the root, which are always different.[2] The unbounded variations of the carbonate alkalinity term when one of the individual species was used instead of $C_T$ furthermore required to modify the stopping criterion for the iterations: in SolveSAPHE v1 iterations are stopped as soon as the relative difference between successive iterates falls below a set tolerance $\epsilon$ ($\epsilon = 10^{-8}$ by default). However, iterations for $Alk_T$ & $CO_2$ and for the greater root of $Alk_T$ & $CO_3^{2-}$ were prone to early termination with that stopping criterion, as iterates only slowly changed due to the extreme gradients in the $Alk_C$ term of the equation. The stopping criterion is therefore now based upon the width of the bracketing interval and iterations are stopped as soon as $(H_{max} - H_{min}) < \epsilon \frac{1}{2}(H_{max} + H_{min})$, where $H_{max}$ and $H_{min}$

---

[2]Both corrections have been backported to the version 1 branch of SolveSAPHE and are included in v1.1 in the SolveSAPHE archive on Zenodo (Munhoven, 2013–2021).

are resp. the upper and lower brackets of the root, which are continuously updated as iterations progress. As a consequence of this change, the number of bisection steps considerably increased. In order to speed up convergence, most bisection steps were replaced by regula falsi steps on $[H_{min}, H_{max}]$. Bisection steps are only used occasionally when either the minimum or maximum root bracket gets updated too often in a row (three times by default) which indicates that the equation values at $H_{max}$ and $H_{min}$ have strongly different magnitudes. Unfortunately, the number of iterations required for the original SolveSAPHE pair $Alk_T$ & $C_T$ increase with this stopping criterion, without any appreciable gain in precision (compare, e.g., the number of iterations from Fig. 3b and the residuals from Fig. 1d in Munhoven (2013), with the number of iterations required here as reported in Fig. 6 for SW3 and the synthetic overview of the equation residuals reported in Tables S4 and S5 in the *Additional Results* in the Supplement). For modelling purposes, where $Alk_T$ & $C_T$ is generally the relevant pair of data, SolveSAPHE v1 remains the most efficient choice. Tests have shown that the two safe-guarded algorithms from SolveSAPHE v1 typically require 40–45% less computing time than their SolveSAPHE-r2 counterparts.

Finally, as explained above, some $Alk_T$ & $CO_3^{2-}$ combinations require the solution of an auxiliary minimisation problem. For this purpose, Brent's algorithm was implemented into SolveSAPHE (translated to Fortran 90 from the Algol 60 version in Brent (1973, Sect. 5.8), taking into account the author's errata reported on https://maths-people.anu.edu.au/~brent/pub/pub011.html and his modifications to the original algorithm as implemented in https://www.netlib.org/go/fmin.f).

## 3.2 Results and discussion

### 3.2.1 Test case definitions

Results from the three test cases SW1, SW2 and SW3 from Munhoven (2013) were used as starting points to define sets of $Alk_T$-$CO_2$, $Alk_T$-$HCO_3^-$ and $Alk_T$-$CO_3^{2-}$ concentration pairs to drive the test case experiments. Two supplementary cases were added here: BW4 for surface brackish water with $S = 3.5$, and ABW5 (based upon the data of Yao and Millero (1995) for the Framvaren Fjord, Norway) for anoxic brackish water with $Alk_T$ and $C_T$ values seven to nine times higher than in the open ocean, as well as comparatively high alkalinity contributions from phosphates, silicates, sulfides, phosphates and ammonium. For $CO_2$ and $CO_3^{2-}$, which are most conveniently handled on a logarithmic concentration scale, the representative ranges were adapted so that the range endpoints are integer powers of ten. The adopted ranges and scales are reported in Table 1. Each of the SW1, SW2 and SW3 test cases is complemented with three sets of temperature, salinity and pressure conditions for typical environments (surface cold, surface warm and deep cold seawater); for BW4 only one such set for cold surface dilute or brackish water is used ($T = 275.15\,\mathrm{K}$, $S = 3.5$) and for ABW5 one set for subsurface brackish water ($P = 13.5\,\mathrm{bar}$, $S = 22.82$).

For the comparison of the computational requirements for the processing of each set of samples, the adopted $[CO_2]$, $[HCO_3^-]$ or $[CO_3^{2-}]$ distributions are adapted. Although the $[CO_2]$, $[HCO_3^-]$ and $[CO_3^{2-}]$ ranges for each test case reported on Table 1 have been defined on the basis of their respective distributions calculated from the $Alk_T$ & $C_T$ results and although the adopted grids have the same dimensions, they do not cover exactly the same "samples" in any given test case. To overcome that inconsistency, each test experiment for the intercomparison is first carried out with the $Alk_T$ & $C_T$ pair and the results stored. For

the other three pairs, the $p$H distribution obtained with the $\text{Alk}_T$ & $C_T$ pair for the corresponding set of temperature, salinity and pressure is first read in and the corresponding $[CO_2]$, $[HCO_3^-]$ or $[CO_3^{2-}]$ distribution calculated on the underlying $C_T$-

415 $\text{Alk}_T$ grid. The so-obtained arrays of species concentrations are then used to define the set of $\text{Alk}_T$-$CO_2$, $\text{Alk}_T$-$HCO_3^-$ and $\text{Alk}_T$-$CO_3^{2-}$ data pairs for the benchmark calculations. This way the test case experiments for the four different characteristic carbonate system concentrations cover exactly the same set of samples in each test case. These sample sets cannot be represented on rectangular $[CO_2]$-$\text{Alk}_T$, $[HCO_3^-]$-$\text{Alk}_T$ or $[CO_3^{2-}]$-$\text{Alk}_T$ grids, respectively, which is nevertheless irrelevant for the histogram syntheses presented in Figs. 6 and 7. These variants of the test cases are denoted $\text{SW1}_{CT}$, $\text{SW2}_{CT}$, $\text{SW3}_{CT}$, $\text{BW4}_{CT}$

and $\text{ABW5}_{CT}$.

### 3.2.2 Results

While all the test cases have their specific relevance, we are going to focus on SW2 for most of our discussion here. SW2 covers currently observed sea-water samples, thus encompassing SW1, and conditions expected to occur over the next 50,000 years as derived from simulation experiments carried out with MBM-MEDUSA (Munhoven, 2009). A wider selection of results also

for the other cases is presented in the *Additional Results* in the Supplement. $p$H distributions for the SW2 test case are shown in Fig. 5.

The difficulties posed by $\text{Alk}_T$ & $CO_2$ that were at the origin of most of the amendments to the solver algorithms show up in the histograms for the number of iterations required to reach convergence shown in Fig. 6 for `solve_at_general` which uses the hybrid Newton-Raphson–regula falsi–bisection scheme and in Fig. 7 for `solve_at_general_sec` which uses the

430 hybrid secant–regula falsi–bisection scheme. With each one of the two solvers, $\text{Alk}_T$ & $CO_2$ problems require in general more iterations to conclude than the other three pairs. This is especially pronounced with `solve_at_general` (Fig. 6), where a considerable fraction of the $\text{Alk}_T$ & $CO_2$ samples require 45 to 55 and more iterations. In comparison, $\text{Alk}_T$ & $C_T$ samples typically require about four to eight iterations for naturally occurring compositions, and only in some rare instances more than twenty for the extreme SW3. The other pairs range between these two, $\text{Alk}_T$ & $HCO_3^-$ coming closest to $\text{Alk}_T$ & $C_T$. ABW5

shows a few deviations from the other tests cases. Here, solving the $\text{Alk}_T$ & $CO_2$ problem with `solve_at_general` nearly always takes more than 50 iterations, with `solve_at_general` almost always nine. The solution of $\text{Alk}_T$ & $C_T$ for ABW5 with `solve_at_general_sec` takes considerably more iterations than $\text{Alk}_T$ & $CO_3^{2-}$ (the fastest) and $\text{Alk}_T$ & $CO_2$ (the second fastest).

Finally, Figs. 6 and 7 demonstrate the superiority of `solve_at_general_sec` over `solve_at_general`. All in all,

the former requires only one fourth to one half of the number of iterations than the latter, and it produces root approximations characterised by equation residuals that are up to seven orders of magnitude lower than those obtained with the former (see Tables S4 and S5 in the *Additional Results* in the Supplement). ABW5 again presents an exception to this general pattern: `solve_at_general_sec` requires typically about twice as many iterations to solve the $\text{Alk}_T$ & $C_T$ problem than `solve_at_general`.

All these observations are also reflected in the execution times of the two solvers. The Newton-Raphson based solver takes more than five times as much time for the SW2 test case with $\text{Alk}_T$ & $CO_2$ than with $\text{Alk}_T$ & $C_T$; for $\text{Alk}_T$ & $CO_3^{2-}$ it takes

four times as much (for both roots though, including the solution of the minimization problem for part of the domain). For $Alk_T$ & $HCO_3^-$, the difference is only 20%. With the secant based method, the picture is completely different: $Alk_T$ & $CO_2$ takes only about 30% more time than $Alk_T$ & $C_T$, $Alk_T$ & $CO_3^{2-}$ twice as much, whereas $Alk_T$ & $HCO_3^-$ executes even about 5% faster. For the $Alk_T$ & $CO_2$ pair of input data the difference between the two solvers is greatest: the secant based one takes less than one fourth of the time taken by the Newton-Raphson based one.

Another key factor that influences the execution times is the initialisation scheme, although the comparisons are not as clear-cut as in Munhoven (2013). Safe initialisation with the geometric mean of the root brackets (the fall-back initialisation value mentioned in Sect. 2.5) results in 40–60% increases of the execution times for the $Alk_T$ & $C_T$ and the $Alk_T$ & $HCO_3^-$ input pairs, compared to the standard cubic polynomial one. Similar increases are obtained with a constant uniform $pH = 8$ initialisation. For $Alk_T$ & $CO_2$ and $Alk_T$ & $CO_3^{2-}$, the differences are much smaller and range between a decrease or an increase of up to 5%. With these two, the quality of the root brackets seems to be more critical than the initial value.

In the analysis in Sect. 2.4.1, two characteristic thresholds for $Alk_T$ have been made out for $\gamma > 0$: an upper one at $L_{\min} + Alk_{nWC}(H_{\min})$, above which the problem always has two $[H^+]$ solutions, and a lower one at $L_{\min} + Alk_{nWCinf}$, below which the problem does not have any solution at all. For intermediate values of $Alk_T$ it is necessary to determine $H_{\tan}$ and $Alk_{\tan}$ to find out how many roots the problem has, and, in case there are two, where the separation between them lies. The minimisation procedure required to determine $H_{\tan}$ is computationally expensive as can be seen in Fig. 8 (for SW2-sc). The most probable number of iterations is in all experiments between 21 and 25; the median number is each time $0.9 \pm 0.5$ higher than the most probable number, due to the skew-symmetric nature of the distribution of the number of iterates, as illustrated in the insert in Fig. 8 (see also Fig. S23 in the *Additional Results* in the Supplement). The subsequent computation of the roots is much cheaper: for the lower root, the secant based algorithm most probably takes five iterations, and only occasionally 15–16, and for the greater root, most probably four and only rarely more than nine. The total number of samples in the SW2 test case is 1.95 million. 10,500 (0.54%) of these do not have any root for the $Alk_T$ & $CO_3^{2-}$ pair and the solution of the minimisation problem is required for 173,445 samples (8.89%), because $H_{\tan}$ is required to separate the two roots. The lower threshold essentially turns out as useless: it ranges at about $-28\,\mathrm{mmol\,kg^{-1}}$. This is due to the hydrogen sulfate acid system which strongly dominates the $Alk_{nWC}$ minimum in seawater, because of the high total sulfate concentration in sea-water ($S_T \simeq 28\,\mathrm{mmol\,kg^{-1}}$). For carbonate ion concentrations below $400\,\mathrm{\mu mol\,kg^{-1}}$, i.e., for most of the naturally occurring waters, the $Alk_T$ & $CO_3^{2-}$ problem will always have two roots and the solution of the auxiliary minimisation problem is not required to characterise them.

## 4 Conclusions

The approach adopted in SolveSAPHE (Munhoven, 2013) to safely determine carbonate speciation in particular, and speciation calculations of mixtures of acids in aqueous solution in general, knowing only the total concentrations of the different acid systems and the total alkalinity of the system was adapted and extended here to use $[CO_2]$, $[HCO_3^-]$ and $[CO_3^{2-}]$ instead of the total inorganic carbon concentration, $C_T$. The rationale can be entirely transposed to these three pairs: (1) the amended

alkalinity-$p$H equations for $Alk_T$ & $CO_2$ and for $Alk_T$ & $HCO_3^-$ still have one and only one positive solution while $Alk_T$ & $CO_3^{2-}$ may have no solution, or one or two; (2) intrinsic brackets that only depend on a priori available information can be derived for the root of the $Alk_T$ & $CO_2$ and $Alk_T$ & $HCO_3^-$ problems, as well as for the two roots of $Alk_T$ & $CO_3^{2-}$ problems that may have to be solved for naturally occurring sample compositions. More uncommon but physically realistic $Alk_T$ & $CO_3^{2-}$ problems may additionally require the solution of an auxiliary minimisation problem to determine the threshold $Alk_T$

value below which the problem does not have any roots and above which it has two of them. The solution of this problem also provides a separation value of the two roots. To our best knowledge, SolveSAPHE is the first package to offer a complete solution of the $Alk_T$ & $CO_3^{2-}$ problem, autonomous above all.

    The two safeguarded numerical solvers from SolveSAPHE v1 have been adapted to allow for the solution of problems that may have up to two roots. The Newton-Raphson–bisection based solver required extensive modifications for the reliable solu-

tion of the numerically far more challenging $Alk_T$ & $CO_2$, $Alk_T$ & $HCO_3^-$ and $Alk_T$ & $CO_3^{2-}$ problems. Most bisection steps have been replaced by regula falsi steps for increased convergence speed. The secant–bisection solver only required minimal adaptations. A Fortran 90 reference implementation, SolveSAPHE-r2, was prepared and used to evaluate the performances of the different methods for solving four benchmark problems. While the secant–bisection method was already slightly superior to the Newton-Raphson–bisection method in SolveSAPHE v1, that advantage has now become overwhelming: in SolveSAPHE-

r2, it typically requires two to four times less iterations, and for the newly handled pairs, the equation residuals are orders of magnitude lower than the Newton-Raphson–regula falsi–bisection based solver (typically of the order of $10^{-19} - 10^{-18}$ compared to $10^{-13} - 10^{-12}$).

    For carbonate speciation problems posed by $Alk_T$ and either one of $[CO_2]$, $[HCO_3^-]$ or $[CO_3^{2-}]$ the secant based routine from SolveSAPHE-r2, `solve_at_general2_sec`, is thus clearly the method of choice; for calculations on the basis of $Alk_T$ &

$C_T$, both `solve_at_general` and `solve_at_general_sec` from SolveSAPHE v1 will perform better, although the secant based solver is marginally faster, once again.

*Code availability.* All the Fortran 90 codes of SolveSAPHE version 1 series (of which v1.0.3 was used to derive the results presented in Fig. 1) are available on Zenodo from Munhoven (2013–2021) for use under the GNU Lesser General Public Licence version 3 (LGPLv3) or later. The codes for SolveSAPHE-r2 (v2.0.1) that are described in this manuscript are included in the Supplement and made available for

use under the same licence. They are also archived on Zenodo (Munhoven, 2021). Future bug-fix releases and updates will also be archived there.

    Epitalon et al. (2021) have ported SolveSAPHE-r2 to R (not used here) for usage under the GNU General Public License version 2 (GPL-2) or 3 (GPL-3).

*Author contributions.* GM did the mathematical analyses presented here, developed all the codes, carried out the numerical experiments,

processed the results, produced the graphs and wrote the paper.

*Competing interests.* The author declares that there is no conflict of interest.

*Acknowledgements.* James Orr provided the kick-off momentum for me to reconsider SolveSAPHE and to complete it in order to combine $Alk_T$ with the individual carbonate system species instead of $C_T$ – thank you. Thanks are also due to Jean-Pierre Gattuso for pointing out issues with the colour schemes adopted in the originally submitted version of the manuscript and for guidance about how to address them. The comments and recommendations by the two anonymous referees were most helpful to improve the framing of the paper and the scope of the results. The analysis based upon Fig. 4 was suggested by Anonymous Referee #1 and recommended for inclusion in the manuscript by Anonymous Referee #2. Financial support for this work was provided by the Belgian Fund for Scientific Research – F.R.S.-FNRS (project SERENATA, grant CDR J.0123.19). GM is a Research Associate with the Belgian Fund for Scientific Research – F.R.S.-FNRS.

## Appendix A: The direct cases

For the sake of completeness, I provide here succinct "recipes" to calculate all the different carbonate system related variables, knowing two of them. Many of these were already known in the 1960s (see, e.g., Park (1969)). The *Guide to Best Practices for Ocean CO₂ Measurements* (Dickson et al., 2007) lists the most commonly used pairs and furthermore includes procedures for selected triplets and quartets, for which not all of the equilibrium constants are required. In the following, we assume that there are direct and invertible relationships between $[CO_2]$ and the fugacity ($f CO_2$) or the partial pressure ($p CO_2$) of $CO_2$ and between $pH$ and $[H^+]$ on any chosen $pH$ scale. We therefore restrict ourselves to $[CO_2]$ and $[H^+]$.

The conditions for the existence of a solution are generally that the concentrations of $H^+$ and of the DIC species are strictly positive. In some instances, the input data must fulfil additional constraints that are, however, not always straightforward to quantitatively state a priori.

**$C_T$ & $CO_2$, $C_T$ & $CO_3^{2-}$** —  (1) With these two pairs, the $[CO_2]/C_T$ fraction, resp. the $[CO_3^{2-}]/C_T$ fraction, is fixed and Eq. (4), resp. Eq. (6), defines a quadratic equation in $[H^+]$ that always allows for exactly one positive solution; (2) calculate the remaining two species concentrations from their respective species fraction; (3) $Alk_T$ from Eq. (1). In addition to the positivity of the species concentrations, the following constraints must be met: $[CO_2] < C_T$ and $[CO_3^{2-}] < C_T$.

**$C_T$ & $HCO_3^-$** —  With this pair, the $[HCO_3^-]/C_T$ fraction, denoted by $b$ hereafter, is fixed and Eq. (5) becomes a quadratic equation in $[H^+]$. That equation has two positive solutions if $b < 1/(1 + 2\sqrt{K_2/K_1})$, one double root if $b = 1/(1 + 2\sqrt{K_2/K_1})$ and no real solutions if $b > 1/(1 + 2\sqrt{K_2/K_1})$. This is well illustrated in Fig. 1b above: there are $C_T$ & $[HCO_3^-]$ combinations that allow for two different $Alk_T$ values and, equivalently, two $pH$ values, and there are others that do not allow for any. Please notice that the threshold fraction $1/(1 + 2\sqrt{K_2/K_1})$ is always lower than 1 and the natural a priori constraint requiring that $b < 1$ is thus insufficient to guarantee a solution: for $T = 275.15\,\mathrm{K}$, $S = 35$ and $P = 0\,\mathrm{bar}$, the threshold ratio is 94.48%.

When there are two roots, one faces a similar dilemma as with the $Alk_T$ & $CO_3^{2-}$ problem: which one to chose? Most often the lower of the two will again be the appropriate one, as that one typically leads to $Alk_T > C_T$, whereas the greater one leads to $Alk_T < C_T$. This criterion might be sufficient to discriminate between the two – in seawater it generally is – but in some instances additional information, quantitative or qualitative, might be of order.

In general: (1) solve the quadratic equation and chose the appropriate of the two roots. (2) Calculate $[CO_2]$ and $[CO_3^{2-}]$ from their respective species fractions; (3) $Alk_T$ from Eq. (1).

**$CO_2$ & $HCO_3^-$** —   (1) $[H^+]$ from $K_1$; (2) $[CO_3^{2-}]$ from $K_2$; (3) $C_T$ can be calculated from the three carbonate species concentrations; (4) $Alk_T$ from Eq. (1).

**$CO_2$ & $CO_3^{2-}$** —   (1) $[HCO_3^-]$ from $[HCO_3^-]^2 = K_1/K_2[CO_2][CO_3^{2-}]$; (2) $C_T$ from the three carbonate species concen-
trations; (3) $[H^+]$ from $K_1$ or $K_2$; (4) $Alk_T$ from Eq. (1).

**$HCO_3^-$ & $CO_3^{2-}$** —   (1) Calculate $[H^+]$ from $K_2$; (2) $[CO_2]$ from $K_1$; (3) $C_T$ from the three carbonate species concentrations; (4) $Alk_T$ from Eq. (1).

**$CO_2$ & $H^+$** —   (1) calculate $[HCO_3^-]$ from $K_1$; (2) calculate $[CO_3^{2-}]$ from $K_2$; (3) $C_T$ from the three carbonate species concentrations; (4) $Alk_T$ from Eq. (1).

**$HCO_3^-$ & $H^+$** —   (1) calculate $[CO_2]$ from $K_1$; (2) calculate $[CO_3^{2-}]$ from $K_2$; (3) $C_T$ from the three carbonate species concentrations; (4) $Alk_T$ from Eq. (1).

**$CO_3^{2-}$ & $H^+$** —   (1) calculate $[HCO_3^-]$ from $K_2$; (2) calculate $[CO_2]$ from $K_1$; (3) $C_T$ from the three carbonate species concentrations; (4) $Alk_T$ from Eq. (1).

**$Alk_T$ & $H^+$** —   (1) $C_T$ from Eq. (2); (2) individual species concentrations from the species fractions.

As illustrated in Fig. 1d–f above, there are $Alk_T$-$[H^+]$ combinations that lead to physically unrealistic negative $Alk_C$. Following Eq. (3) negative $Alk_C$ requires negative $C_T$ and vice-versa.

The shape of the blank area depends on the non-carbonate contributors to the total alkalinity. In practice, such incompatible combinations are unlikely to arise from measurements, except if the adopted set of $Alk_T$ contributors is inappropriate.

**$C_T$ & $H^+$** —   Individual species concentrations from the species fractions; $Alk_T$ from Eq. (1).

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

**Table 1.** Ranges of variation for the input variables for the five test cases. Experiments always considered $\text{Alk}_\text{T}$ and either one of $C_\text{T}$, $[CO_2]$, $[HCO_3^-]$ or $[CO_3^{2-}]$.

| | | SW1 | | SW2 | | SW3 | | BW4 | | ABW5 | |
| --- | --- | --- | --- | --- | --- | --- | --- | --- | --- | --- | --- |
| | scale | min | max | min | max | min | max | min | max | min | max |
| $\text{Alk}_\text{T}/[\text{mmol kg}^{-1}]$ | linear | 2.20 | 2.50 | 2.20 | 3.50 | $-1.0$ | 5.0 | 0.0 | 1.5 | 17.0 | 20.0 |
| $C_\text{T}/[\text{mmol kg}^{-1}]$ | linear | 1.85 | 2.45 | 1.85 | 3.35 | 0.0 | 6.0 | 0.0 | 1.2 | 15.0 | 17.5 |
| $[CO_2]/[\text{mol/kg}]$ | log. | $10^{-6}$ | $10^{-3}$ | $10^{-7}$ | $10^{-3}$ | $10^{-14}$ | $10^{-2}$ | $10^{-12}$ | $10^{-3}$ | $10^{-4}$ | $10^{-2}$ |
| $[HCO_3^-]/[\text{mmol/kg}]$ | linear | 1.20 | 2.40 | 0.60 | 3.20 | 0.0 | 5.0 | 0.0 | 1.0 | 13.0 | 17.0 |
| $[CO_3^{2-}]/[\text{mol/kg}]$ | log. | $10^{-5}$ | $10^{-3}$ | $10^{-6}$ | $10^{-3}$ | $10^{-14}$ | $10^{-2}$ | $10^{-9}$ | $10^{-3}$ | $10^{-5}$ | $10^{-3}$ |

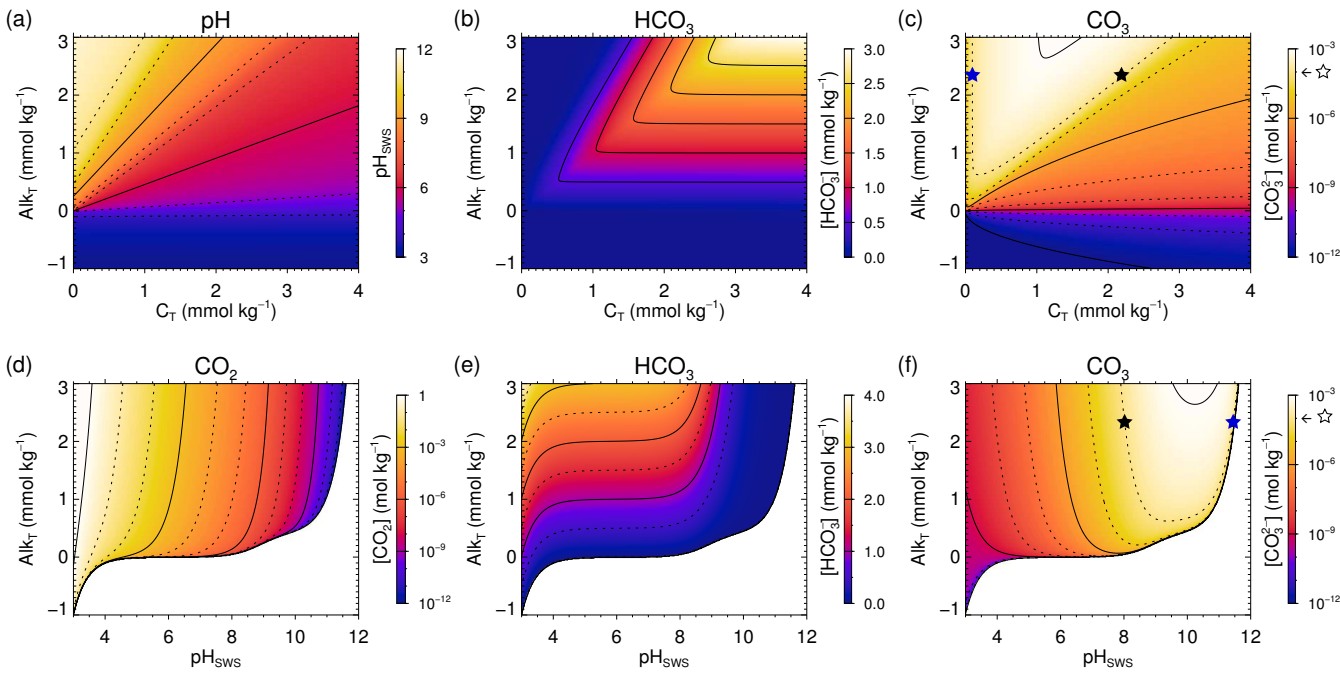

**Figure 1.** (a) $p$H; (b) $HCO_3^-$ and (c) $CO_3^{2-}$ concentration isolines in $C_T$-$Alk_T$ space; (d) $CO_2$, (e) $HCO_3^-$ and (f) $CO_3^{2-}$ concentration isolines in $p$H-$Alk_T$ space. Blank areas in ((d), (e) and (f) represent the $p$H-$Alk_T$ combinations that lead to negative $Alk_C$. The blue and the black stars in (c) and (f) locate the two possible $C_T$ and $p$H roots for a sample with $Alk_T = 2.3\,\mathrm{mmol\,kg^{-1}}$ and $[CO_3^{2-}] = 0.1\,\mathrm{mmol\,kg^{-1}}$ (one of the dashed isolines, as indicated by the open star symbol in the colour scale). Figure 3 in Deffeyes (1965) is similar to (c).

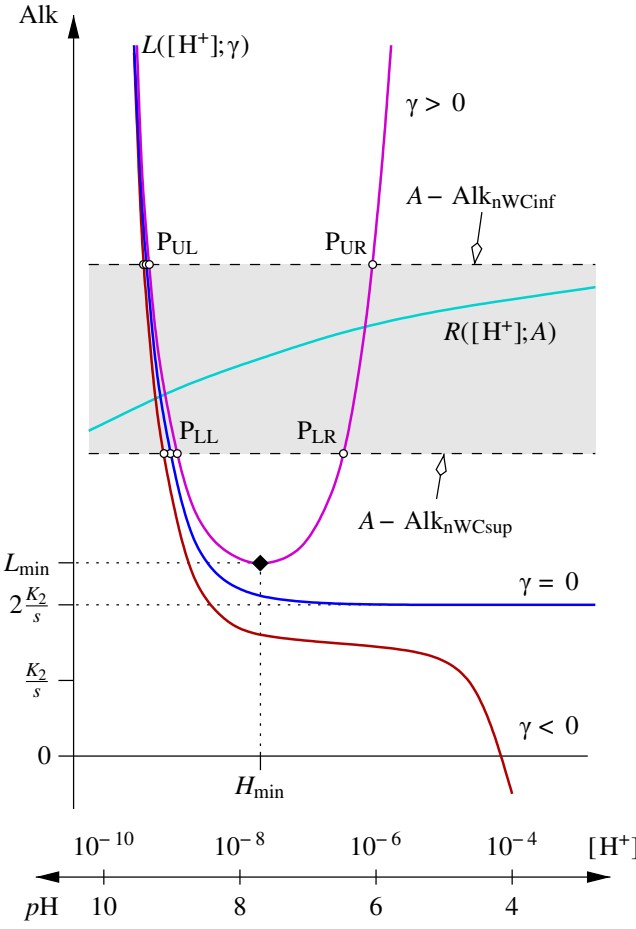

**Figure 2.** Schematic representation of the general characteristics of the $L([H^+]; \gamma)$ and $R([H^+]; A)$ components of the alkalinity-$p$H equation for the $Alk_T - CO_3^{2-}$ pair. The grey band delimits the (monotonous) variations of $R([H^+]; A)$, for a given alkalinity $A$. The band moves up and down without being distorted as $A$ is increased, resp., decreased. For a given pair of $Alk_T$ and $CO_3^{2-}$ concentrations, the actual equation to solve is $L([H^+]; \gamma) = R([H^+]; Alk_T)$, where $\gamma = \frac{[CO_3^{2-}]}{K_2} - \frac{1}{s}$. $\gamma = 0$ thus corresponds to $[CO_3^{2-}] = \frac{K_2}{s}$.

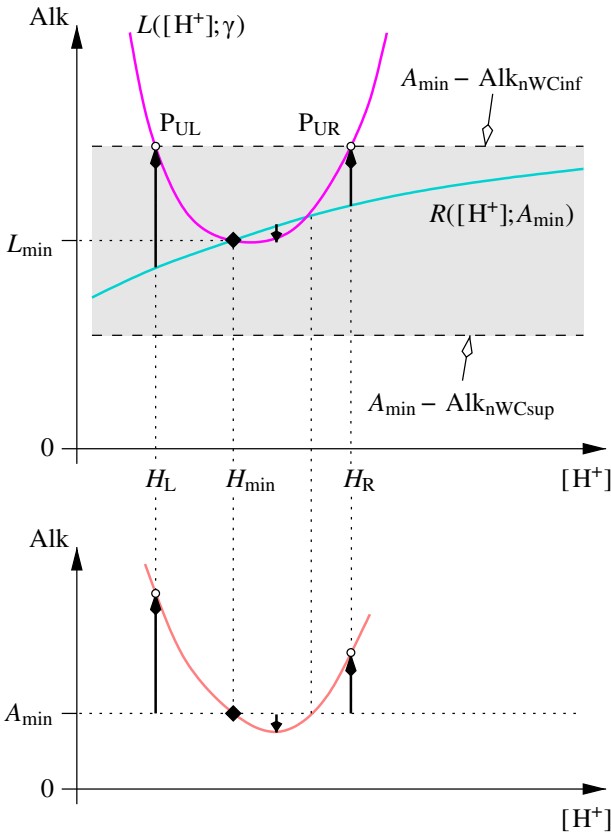

**Figure 3.** Determination of the $A$ value for which the $L([H^+]; \gamma)$ and $R([H^+]; A)$ curves become tangent, or, equivalently, the lowest $Alk_T$ value for which the equation $L([H^+]; \gamma) - R([H^+]; Alk_T) = 0$ has a solution. The top panel shows how relevant characteristic points can be derived by considering the particular $R([H^+]; A)$ curve that intersects $L([H^+]; \gamma)$ at its minimum. The bottom panel shows the locus of the solutions of $L([H^+]; \gamma) - R([H^+]; Alk_T) = 0$ in an $[H^+] - Alk_T)$ graph, i.e., the curve $Alk_T = L([H^+]; \gamma) + Alk_{ncW}([H^+])$. Please notice that $A_{min} = L_{min} + Alk_{ncW}(H_{min})$ denotes the alkalinity value obtained for $[H^+] = H_{min}$, and not the minimum value of the curve shown on the bottom panel. See text for details.

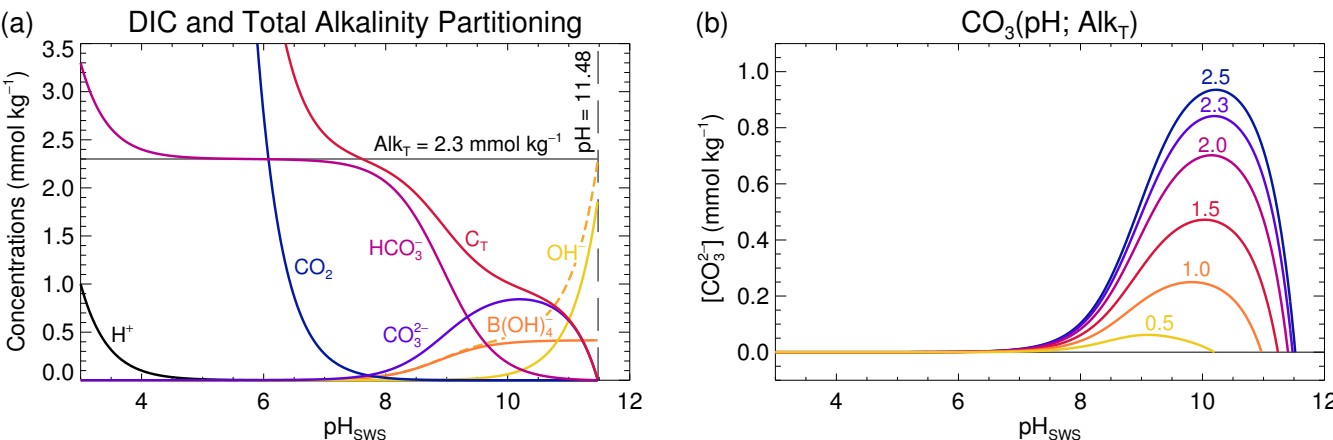

**Figure 4.** (a) Evolutions of the concentrations of the different species composing $Alk_T$ and $C_T$, as a function of $p$H, for $Alk_T = 2.3\,\text{mmol}\,\text{kg}^{-1}$. $C_T$ and the concentrations all of its components reduce to 0 at $p\text{H} = 11.48$ (marked by the long-dashed vertical black line) in this example. The dashed orange line represents the joint contribution of $B(OH)_4^-$ and $OH^-$ which are the dominant $Alk_T$ contributors at high $p$H. (b) $[CO_3^{2-}]$ as a function of $p$H for different $Alk_T$ values (indicated in $\text{mmol}\,\text{kg}^{-1}$ for each curve). Each curve represents a horizontal cross-section at the corresponding $Alk_T$ level through the $[CO_3^{2-}]$ distribution depicted in Fig. 1e.

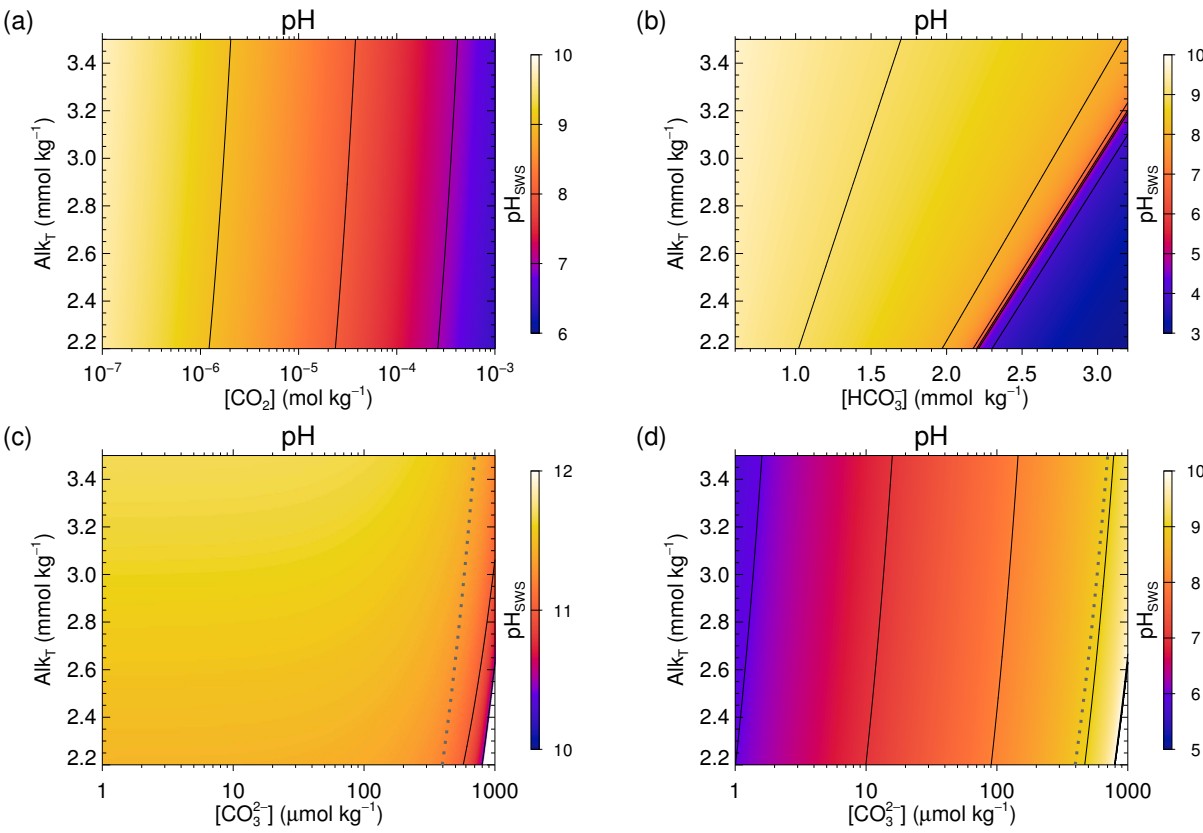

**Figure 5.** $p$H distributions for the SW2-sc test case (SW2 under cold surface conditions, where $T = 273.15\,\mathrm{K}$, $S = 35$ and $P = 0\,\mathrm{bar}$), obtained with `solve_at_general2_sec`: (a) $\mathrm{Alk_T}$ & $CO_2$; (b) $\mathrm{Alk_T}$ & $HCO_3^-$; (c) the lower $[H^+]$ root (higher $p$H root) of $\mathrm{Alk_T}$ & $CO_3^{2-}$; (d) the greater $[H^+]$ root (lower $p$H root) of $\mathrm{Alk_T}$ & $CO_3^{2-}$. The thick grey dashed line in (c) and (d) shows the critical limit above which the $\mathrm{Alk_T}$ & $CO_3^{2-}$ always has two roots. Below this limit further calculations are required to determine the number of solutions. More details are given in the text and in the Supplement. Please notice the different scales on the horizontal axes and for the $p$H colour coding in the four panels.

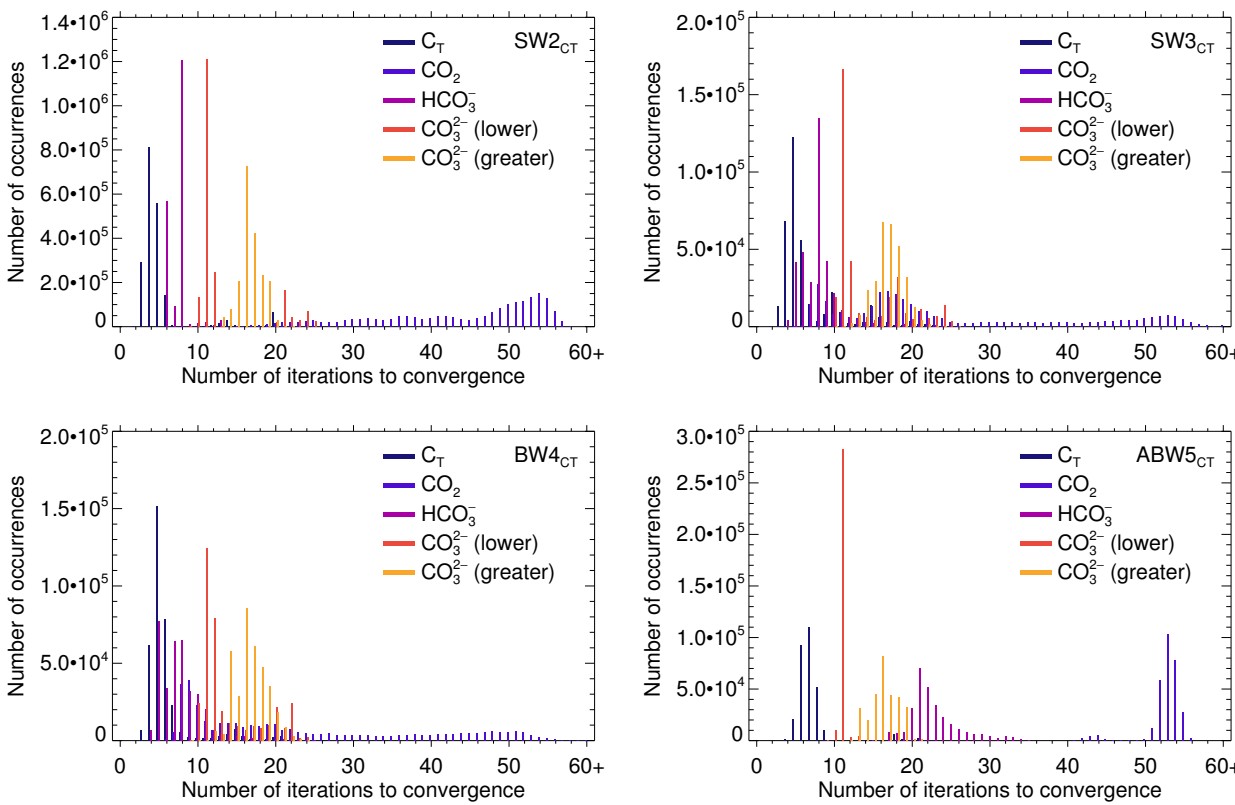

**Figure 6.** Number of iterations to convergence required by the various data pairs (separately for the lower and the greater $[H^+]$ roots of the $Alk_T$ & $CO_3^{2-}$), for each of the four test cases, carried out with `solve_at_general` (using a hybrid Newton-Raphson–regula falsi–bisection method). The absolute maximum numbers of iterations were 58, 67, 64 and 56, for SW2, SW3, BW4 and ABW5, resp., and 58 for SW1 (not shown).

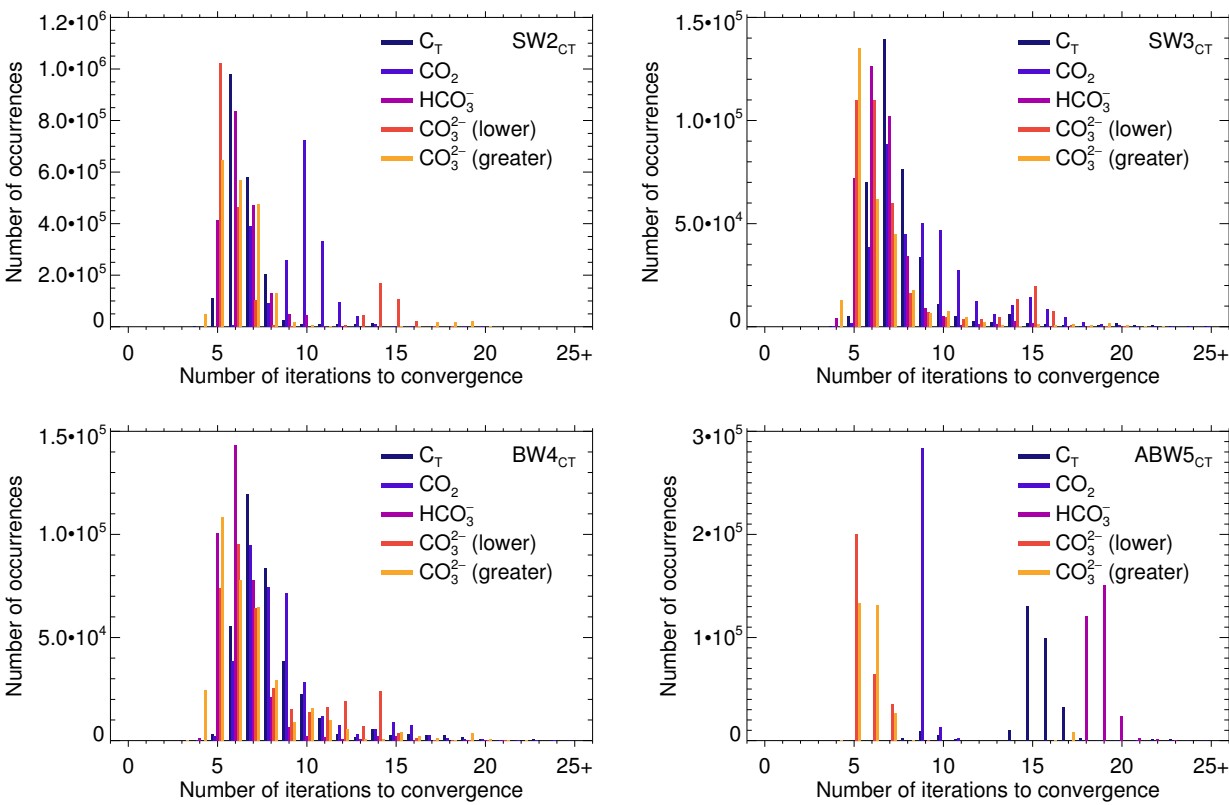

**Figure 7.** Number of iterations to convergence required by the various data pairs (separately for the lower and the greater $[H^+]$ roots of the $\text{Alk}_T$ & $CO_3^{2-}$), for each of the four test cases, carried out with `solve_at_general_sec` (using a hybrid secant–regula falsi–bisection method). The absolute maximum numbers of iterations were respectively 20, 21, 29 and 27, for SW2, SW3, BW4 and ABW5, resp., and 20 for SW1 (not shown).

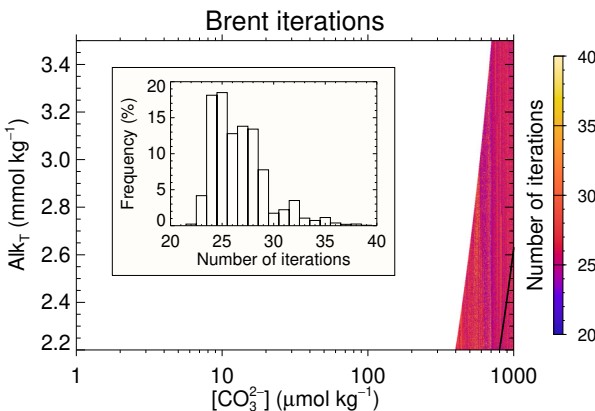

**Figure 8.** Number of iterations required by Brent's algorithm in the SW2 test case to solve the auxiliary minimisation problem whose solution determines the number of roots of the $Alk_T$ & $CO_3^{2-}$ pair and also provides the separation between the two roots. The white area covers the region where the solution of the minimisation problems is not required as $Alk_T$ is sufficiently high so that there were two roots. The insert shows the frequency distribution of the number of iterations required. The black line in the lower right corner traces the limit between regions with two roots and without roots (compare with Figs. 5c and 5d).