# Peer review of "SolveSAPHE-r2 (v2.0.1): revisiting and extending the Solver Suite for Alkalinity-PH Equations for usage with $CO_2$ , $HCO_3^-$ or $CO_3^{2-}$ input data"

_Geoscientific Model Development, 2020_

## Referee Comment (RC1)

**Review on SolveSAPHE-r2: revisiting and extending the Solver Suite for Alkalinity-pH Equations for usage with $CO_2$, $HCO_3^-$ or $CO_3^{2-}$ input data by Guy Munhoven**

Guy Munhoven has investigated the number of solutions (pH values) of the carbonate system for the pairs $Alk_T$-$CO_2$, $Alk_T$-$HCO_3^-$, and $Alk_T$-$CO_3^{2-}$. Using detailed mathematical analysis he has shown that for reasonable values for the variables in the first two pairs one obtains a unique (one and only) physically sensible solution (positive concentration of all compounds). In contrast, the pair $Alk_T$-$CO_3^{2-}$ is more of a challenge: (1) Depending on the concentration of $CO_3^{2-}$ three cases have to been discerned; although all three cases have been properly analyzed by Munhoven, the third one (with $\gamma > 0$) is the most relevant (the only relevant in the ocean). (2) For $CO_3^{2-}$ concentrations $> 1$ $\mu$mol kg$^{-1}$ (implying $\gamma > 0$) either two or no physically sensible solution exists.

Based on his detailed mathematical insight, Munhoven has developed algorithms to solve the various equations (in order to be efficient, good bounds and initial values are useful). The algorithms were coded in FORTRAN-90 and extensively tested.

After Munhoven (2013) this is another outstanding contribution by Guy Munhoven. It is clearly written, however, not 'easy reading'. With respect to mathematical analysis, the work can be considered as the capstone (a heavy one!) to the arc of all 15 possible pairs of carbonate system parameters.

I have only one major comment (which is beyond mathematics) and a few minor points.

When calculating pH from alkalinity and carbonate ion concentration in typical seawater (let's say at $Alk_T$ = 2300 $\mu$mol kg$^{-1}$ and 100 $\mu$mol kg$^{-1}$ $<$ [$CO_3^{2-}$] $< 600$ $\mu$mol kg$^{-1}$) one obtains two solutions and the question is 'Which one to take?'. In order to shed some light on this problem I have calculated the concentrations of all carbonate system parameters for a fixed alkalinity ($Alk_T$ = 2300 $\mu$mol kg$^{-1}$) as a function of pH (Fig. 1). For [$CO_3^{2-}$] = 500 $\mu$mol kg$^{-1}$ one obtains two solutions: $pH_1$ = 8.70 and

$pH_2$ = 10.75 with quite different DIC values. The higher pH value is discarded because the DIC is 'unrealistically' low and that pH = 8.70 would be the chosen answer. This chioce has been taken, for example, by Zeebe & Wolf-Gladrow (2001, p. 277, 'use the larger one'): they recommend using the real solution with the largest $H^+$ concentration (lowest pH). Although this choice works out fine in common seawater, it has not been properly justified. And the justification is actually beyond mathematics.

[Figure]

Figure 1: Illustration of varying number of solution for given TA (black dash-dotted line; TA = 2300 $\mu$mol kg$^{-1}$ is a typical/characteristic oceanic value)) and carbonate ion concentration, [CO$_3^{2-}$]. The results are plotted over pH (surface ocean pH ist slightly above 8, lower values can be found in deeper layers, higher values in the surface during algal blooms (up to 9) or in sea ice brines (above 10, however, at different TA values). At pH = 7 most of TA is in the form of bicarbonate whereas [CO$_3^{2-}$] is relative small. Above pH = pK$_2$ (here = 9.1) the concentration of CO$_3^{2-}$ is larger than that of HCO$_3^-$ and clearly dominates the contribution to TA. However, with further increasing pH the concentration of OH$^-$ becomes larger and larger and at pH $\approx$ 11 is dominating by far the contribution to TA; this is only possible when the concentrations of bicarbonate and carbonate ions go down. Thus whereas the pair TA-CO$_3^{2-}$ possesses always two solutions at pH < 11 (as an example: for TA = 2300 $\mu$mol kg$^{-1}$ and [CO$_3^{2-}$] = 500 $\mu$mol kg$^{-1}$ pH = 8.70 and pH = 10.75 are solutions) there does not exist a solution at pH > 11.

**Minor points**:

Units of alkalinity: I suggest replacing meq (outdated) by mmol (compare, for example, Dickson et al., 2007, Chapter 5, Table 2)

L50 typo: whoe $\rightarrow$ whose

L51 $'s$ is a factor to convert from that scale to the free scale': it might be useful to mention that the value of $s$ is close to 1

Fig. 1: axes labels (quantities & units) missing, same for color bars; remove titles (numbers); $y$-axes from 1 to 0 to 3 or from -1 to 3 (???); a bit more explanation/discussion might be in order

L174 typo: eq. (12then $\rightarrow$ eq. (12) then

L193 $[H^+] \gg$: something missing here

L210 $H_1 < H_{\min}$ and $H_2 > H_{\min}$ might be shortened to $H_1 < H_{\min} < H_2$

L229 'exact knowledge determination' ???

L271 API = ???

L275 'In the course of the development s related to ...' ??? something missing here?

L291 'equation function' ???

Fig. legend: 'two roots The' dot missing after roots

**Sup. Mathematical and Technical Details**

Typos:

2.3.1 $B(OH_3) \rightarrow B(OH)_3$

2.3.3 $H_3(PO)_4 \rightarrow H_3PO_4$

2.3.4 $H_4(SiO)_4 \rightarrow H_4SiO_4$

**MATLAB code for Fig. 1:**

```
K1 = 1.1e-6; K2 = 7.8e-10; KW = 2.3e-14; KB = 1.9e-9; % use rounded values
bor = 0;  % contribution by borate ignored
alk = 2300e-6; % (mol/kg) typical sea water value
pHa = 7:0.1:12; L = length(pHa);
for i = 1:L
    pH = pHa(i); h = 10^(-pH);
    % Zeebe & Wolf-Gladrow (2001):
s = (alk-KW./h+h-KB*bor./(KB+h)) / (K1./h+2.*K1*K2./h./h); % [CO2]
dic = s*(1.+K1/h+K1*K2./h./h);
hco3 = dic./(1+h./K1+K2./h);
co3 = dic./(1+h/K2+h*h/K1/K2);
CO2(i) = s; HCO3(i) = hco3; CO3(i) = co3; DIC(i) = dic;
Hp(i) = h; OH(i) = KW/h;
end
% 2 solutions for a single CO3: example
CO3ex = 500e-6;
% ------------ CO3 and ALK given ------------------
co3 = CO3ex;
p5  = -co3/K2+1.;
p4  = alk - co3*(K1/K2+(KB+2.*K2)/K2) + KB + K1;
p3  = alk*(KB+K1)-co3*(K1+K1*(KB+2.*K2)/K2+2.*KB) ...
    + (-KB*bor-KW+K1*KB+K1*K2);
p2  = alk*(KB*K1+K1*K2)-co3*(K1*(KB+2.*K2)+2.*KB*K1) ...
    + (-KW*KB-K1*KB*bor-K1*KW+K1*K2*KB);
p1  = alk*KB*K1*K2-co3*2.*KB*K1*K2-K1*KW*KB ...
    + (-K1*K2*KB*bor-K1*K2*KW);
p0  = -K1*K2*KW*KB;
p   = [p5 p4 p3 p2 p1 p0];
r   = roots(p);
% 2 real solutions?
isol = 0;
for j = 1:5
   if (real(r(j)) > 0) \&\& (imag(r(j)) == 0)
     isol = isol+1; pHsol(isol) = round(-log10(r(j)),2)
   end
end
```

```
xp = [min(pHa) max(pHa)]; yp = [alk alk];
plot(pHa,DIC,'b',xp,yp,'k-.',pHa,HCO3,'r--',pHa,CO3,'k', ...
    pHa,CO2,'m:',pHa,OH,'c:', ...
    pHsol(1),CO3ex,'k*',pHsol(2),CO3ex,'k*','LineWidth',2)
text(9.8,2100e-6,'TA','Color','black','Fontsize',fs)
text(9.8,1300e-6,'DIC','Color','blue','Fontsize',fs)
text(9.8,700e-6,'CO_3^{2-}','Color','black','Fontsize',fs)
text(7.8,1500e-6,'HCO_3^-','Color','red','Fontsize',fs)
text(7.1,300e-6,'CO_2','Color','magenta','Fontsize',fs)
text(11,2100e-6,'OH^-','Color','cyan','Fontsize',fs)
text(8.3,500e-6,[num2str(pHsol(1))],'Color','black','Fontsize',fs)
text(10.8,500e-6,[num2str(pHsol(2))],'Color','black','Fontsize',fs)
axis([min(pHa) max(pHa) 0 max(DIC)])
xlabel('pH','Fontsize',fs)
ylabel('Concentration (mol/kg)','Fontsize',fs)
set(gca,'Fontsize',fs)
% print('-dpng','MunhovenTAoverpH.png')
```

**References**

[1] Dickson, A.G., C.L. Sabine, and J.R. Christian. *Guide to best practices for ocean* $CO_2$ *measurements*. North Pacific Marine Science Organization, 2007.

[2] Munhoven, Guy. Mathematics of the total alkalinity – pH equation – pathway to robust and universal solution algorithms: the SolveSAPHE package v1. 0.1. *Geoscientific Model Development*, 6(4):1367–1388, 2013.

[3] Zeebe, R.E. and D. Wolf-Gladrow. $CO_2$ in Seawater: Equilibrium, Kinetics, Isotopes: Equilibrium, Kinetics, Isotopes. 346 pp, Elsevier, 2001.

---

## Author Comment (AC1)

[Figure]

**Figure 1.** (a) *pH* isolines; (b) $CO_3^{2-}$ concentration isolines in $C_T$-Alk$_T$ space; (c) $CO_3^{2-}$, (d) $CO_2$ and (e) $HCO_3^-$ concentration isolines in *pH*-Alk$_T$ space. These distributions were calculated with SOLVESAPHE version 1.0.3. For (c), (d) and (e), carbonate alkalinity, Alk$_C$, was derived by using eq. (2), combined with with eqs. (9), (7) and (8) to derive $[CO_3^{-2}]$, $[CO_2]$ and $[HCO_3^-]$, resp. Blank areas represent the *pH*-Alk$_T$ combinations that lead to negative Alk$_C$. Fig. 3 in Deffeyes (1965) is similar to (b).

---

## Author Comment (AC2)

**SolveSAPHE-r2: revisiting and extending the Solver Suite for Alkalinity-PH Equations for usage with $CO_2$, $HCO_3^-$ or $CO_3^{2-}$ input data**

**Reply to the Referees' Comments**

Guy Munhoven

Guy.Munhoven@uliege.be

6th May 2021

I thank both referees for their welcoming reviews and the careful reading of my manuscript, and of its Supplement. I greatly appreciate their constructive, thoughtful and thought provoking comments, ideas and supporting calculations that will be very valuable for revising the manuscript.

**In General ...**

... my reading of the two Anonymous Referees' comments (hereafter resp. AR#1 and AR#2) is that they have essentially three major requests or recommendations to make.

**1 Improve the Framing of the Story**

AR#2 recommends to make"[...] the paper more appealing to a wider audience" and has given pertinent and precise references to the recent literature for this purpose. These references will be valuable to improve the introduction to the theme. The framing of the study will be amended along the lines suggested by AR#2., i. e., by considering $CO_3^{2-}$ as the fifth measurable besides $Alk_T$, $C_T$, $(p)CO_2$ and $pH$. Sharp and Byrne (2019) have shown that $Alk_T$ & $[CO_3^{2-}]$ and $C_T$ & $[CO_3^{2-}]$ are the most suitable data pairs to use for the carbonate system speciation, given the uncertainties of all the measurables and of all the various parameters that enter these calculations. While the $C_T$ & $[CO_3^{2-}]$ problem is straightforward to solve as it only requires the solution of a quadratic equation that always has only one positive root, the $Alk_T$ & $[CO_3^{2-}]$ counterpart is more difficult to address, because of the complications that result from the existence of two physically realistic roots and because it requires an iterative approach. A reliable and fail-safe solution algorithm is therefore of order.

**2 $Alk_T$ & $CO_3^{2-}$: A Tale of Two Solutions**

Both AR#1 and AR#2 were concerned about the question which root shall be chosen when there are two of them?

I first of all thank AR#1 for pointing out the little important detail in the solution recipe for the $Alk_T$ & $CO_3^-$ pair in Zeebe and Wolf-Gladrow (2001, pp. 276–277), that had escaped my attention: "Roots: two positive (use the larger one), three negative." Thank you also for the instructive

calculations and the supporting MATLAB code that allowed to see exactly what simplifications were adopted.

I have been convinced for some time already that the existence of two roots had to do with $CO_3^{2-}$ and $OH^-$ swapping their roles as dominant contributors to $Alk_T$ with increasing $pH$. The calculations and the graph provided in your comment provides a straightforward and simple way to illustrate this.

The observation that the concentration of $CO_3^{2-}$ as a function of $pH$ (I use $[H^+]$ here instead), for a given $Alk_T$ goes to 0 at some $pH$ value is actually universally true. Let us denote the function that describes the evolution of that concentration by $co_3([H^+]; Alk_T)$, to distinguish it from any given $[CO_3^{2-}]$ and also because that function becomes negative above some threshold $pH$, which would be meaningless for $CO_3^{2-}$. We have

$$co_3([H^+]; Alk_T) = \frac{Alk_T - Alk_{nWC}([H^+]) - \frac{K_W}{[H^+]} + \frac{[H^+]}{s}}{\frac{[H^+]}{K_2} + 2}, \tag{1}$$

which is to be understood as a parametric function of $[H^+]$, with $Alk_T$ as a parameter. There are two noteworthy facts about this function:

1. as $[H^+] \to +\infty$, $co_3([H^+]; Alk_T) \to \frac{K_2}{s}$, i. e., the value that $[CO_3^{2-}]$ takes for $\gamma = 0$;

2. as $[H^+] \to 0^+$, $co_3([H^+]; Alk_T) \to -\infty$.

Both limits are independent of $Alk_T$. Accordingly, $co_3([H^+]; Alk_T) > 0$ for sufficiently large $[H^+]$ (i. e., sufficiently low $pH$) on one hand, while $co_3([H^+]; Alk_T) < 0$ for sufficiently low $[H^+]$ (i. e., sufficiently high $pH$) on the other hand. The equation $co_3([H^+]; Alk_T) = 0$ must therefore have at least one root. It actually has exactly one, for any value of $Alk_T$, since this simply requires that the numerator at the right-hand side of the definition of $co_3$ (Eq. (1)) is 0, i. e., that $[H^+]$ is the solution of a standard alkalinity-$pH$ equation where $C_T = 0$. Such an equation always has exactly one positive solution, for any physically meaningful set of total concentrations of the different acid-base systems at play and any $Alk_T$ value (Munhoven, 2013).

In the particular case presented by AR#1, where $Alk_T := [HCO_3^+] + 2[CO_3^{2-}] + [OH^-] - [H^+]$, it is normal that there is no solution possible for $pH > 11$: at sufficiently high $pH$, $OH^-$ becomes the single most important, if not the only significant, contributor to $Alk_T$ as is illustrated on the graph provided. At this point $Alk_T \simeq [OH^-]$, i. e., $Alk_T \simeq K_W/[H^+]$. The corresponding $pH$ value is $pH = -\log(K_W/Alk_T)$. With the values adopted by AR#1 ($K_W = 2.3 \times 10^{-14}\,\mathrm{mol\,kg^{-1}}$ and $Alk_T = 2300\,\mu\mathrm{mol\,(kg\text{-}SW)^{-1}}$, we find that this threshold $pH$ is 11. Not only are there no solutions for $pH > 11$, but $pH$ values above 11 are actually incompatible with $Alk_T$ fixed at $2300\,\mu\mathrm{mol\,(kg\text{-}SW)^{-1}}$. Unlike shown on the graph by AR#1, $[OH^-]$ cannot grow beyond the point where $co_3([H^+]; Alk_T) = 0$ as it is the only contributor to $Alk_T$ beyond that point. With $Alk_T$ is fixed, $[OH^-]$ essentially also becomes fixed in this simple configuration once $[CO_3^{2-}]$ has vanished. So, it is not possible for $pH$ to increase beyond that point.

This is, however, not only true in the simple example proposed by AR#1, but in general and is reflected by the blank areas in Fig. R2, panels (c) to (e), below (Fig. R2 is the revised version of Fig. 1 from the manuscript).

Following the suggestion of AR#2, I have prepared two graphs similar to that of AR#1 that I am going to include in the revised manuscript.

- The first one is equivalent to that of AR#1, except that it is based upon the results presented in Fig. 1 in the submitted manuscript (see Fig. R2). These results also include borate alkalinity (and actually phosphate and ammonia alkalinities as well, which are, however, too low to yield any discernible contribution on the graph and have therefore been omitted). The range of $pH$ values has furthermore been extended to match that of Fig. 1, panels (c)–(e), in the manuscript.

- The second one presents the $CO_3^{2-}$ concentrations as a function of $p$H at fixed $Alk_T$ for different values of $Alk_T$. The different curves are thus horizontal cross-sections through the $[CO_3^{2-}]$ distribution shown in Fig. 1c in the manuscript (Fig. R2e below) at different $Alk_T$ levels.

The two graphs are shown in Fig. R1 in this comment.

There is a noteworthy feature regarding the different $co_3$ curves shown in Fig. R1b. The locus of their maxima actually has a seemingly simple analytical equation:

$$co_{3\,\mathrm{max}}(H) = K_2 \left( -\frac{d\mathrm{Alk}_{\mathrm{nWC}}}{dH} + \frac{K_W}{H^2} + \frac{1}{s} \right) \tag{2}$$

where $H$ is used as a shorthand for $[H^+]$. The derivative of $\mathrm{Alk}_{\mathrm{nWC}}$ is always negative (Munhoven, 2013) and this expression is thus always positive.

At first sight Eq. (2) might appear to offer an alternative to the $A$ minimisation procedure described in the manuscript to determine the number of roots of the problem for a given $Alk_T$ & $CO_3^{2-}$ data pair as it gives direct access to the maximum of a $co_3$ curve as a function of $p$H. Obviously, knowing the characteristics of the maximum of the $co_3$ obtained for a given $Alk_T$ (i. e., the $p$H value that locates the maximum and the maximum value itself) would directly allow to conclude about the number of roots, and, if there are two of them, provide a separation between them. Unfortunately, Eq. (2) somehow has that information only backwards: it provides the maximum value of the curve that has its maximum at a given $p$H value, without knowing which $Alk_T$ that curve corresponds to. To find that $Alk_T$ value one has to invert the function, i. e., to calculate the $p$H at which a given $co_{3\,\mathrm{max}}$ is reached. With those two pieces of information, Eq. (1) would then allow to determine the $Alk_T$ corresponding to the curve.

Notwithstanding the complications related to the inversion of $co_{3\,\mathrm{max}}(H)$ defined by Eq. (2), it should be noticed that minimizing $A([H^+]; [CO_3^{2-}])$ as is done in SOLVESAPHE-r2 and maximizing $co_3([H^+]; Alk_T)$ are actually two completely equivalent problems. Minimising $A([H^+]; [CO_3^{2-}])$ presents one important advantage over maximising $co_3([H^+]; Alk_T)$: negative $A$ values are perfectly acceptable, while they are meaningless when it comes to $co_3$. Appropriate algorithms for the maximisation of $co_3$ would therefore be more complicated to design and implement as they would require additional safeguards.

Remains the burning question:

*Which root chose when there are two of them?*

First of all, it is difficult to say whether the answer to this question is beyond the scope of this paper (AR#2)—may be it is, may be it is not—but I certainly agree with AR#1 that it is "beyond mathematics."

There is no universally valid a priori justification to prefer one of the two solutions over the other and additional information, qualitative or quantitative, will be required to chose. This could be a third measurable, but often even qualitative information only about, say, the expected $p$H or the $C_T$ range might be sufficient. For natural sea- or freshwater samples, it will generally be the lower of the two $p$H solutions that will be the relevant one (in terms of $[H^+]$, the "use the larger one" advice of Zeebe and Wolf-Gladrow (2001)). The high-pH solution generally goes together with $C_T \simeq [CO_3^{2-}]$. For the surface cold conditions at the basis of the results shown in Fig. R2, $CO_3^{2-}$ represents more than 90% of DIC for $p$H $> 10$, as can be calculated from Eq. (6) in the manuscript. From Fig. R2, one can see that $co_{3\,\mathrm{max}} > 0.47\,\mathrm{mmol\,kg^{-1}}$ for $Alk_T \geq 1.5\,\mathrm{mmol\,kg^{-1}}$ and that it is located at $p$H $> 10$. Since the larger of the two solutions is always at greater or equal $p$H than that maximum, we may conclude that for $[CO_3^{2-}] < 0.47\,\mathrm{mmol\,kg^{-1}}$ and $Alk_T \geq 1.5\,\mathrm{mmol\,kg^{-1}}$, the greater of the two $p$H roots, if it exists, always implies that $CO_3^{2-}$ represents more than 90% of DIC. Accordingly, even a rough estimate of one of the other relevant parameters of the carbonate system might be sufficient to reject one of the two roots.

[Figure]

Figure R1: (a) Evolutions of the different species composing $Alk_T$ and $C_T$ as a function of $pH$, for $Alk_T = 2.3 \, \text{mmol kg}^{-1}$. $C_T$ and all of its components reduce to 0 at $pH = 11.48$ (marked by the long-dashed vertical black line) in this example. The short-dashed orange line represents the joint contribution of $B(OH)_4^-$ and $OH^-$ which are the dominant $Alk_T$ contributors at high $pH$. (b) $co_3([H^+]; Alk_T)$ as a function of $pH$ for different $Alk_T$ values (indicated in mmol kg$^{-1}$ for each curve). Each curve represents a horizontal cross-section at the corresponding $Alk_T$ level through the $[CO_3^{2-}]$ distribution depicted in Fig. R2e below.

SOLVESAPHE-r2, which is meant to be universally applicable therefore always determines both roots and leaves it to the user's responsibility to select the relevant one.

In practice, the problem is perhaps not as insurmountable as it might appear at first sight: all the standard procedures for the determination of the total alkalinity of a water sample that I am aware of involve a titration procedure, which requires … $p$H monitoring, and so it should always be possible to get some information about the sample's $p$H, although it might not always be recorded. Depending on the titration procedure adopted (open- or closed-cell – see, e. g., Dickson et al. (2007)), the titration data possibly also allow to get at least an approximate estimate of the sample's $C_T$.

**3 New Test Case: ABW5**

AR#2 suggests to include some other case studies, thinking more specifically about [sediment] pore waters.

I had difficulties to secure sufficiently complete data sets for porewater chemistry to design a realistic representative test case as requested by AR#2. As the purpose was to cover samples where "[…] the concentrations of various acid-base systems may be higher, especially the relative contributions of non-carbonate bases to Alk" I finally resorted to using the data of Yao and Millero (1995) for the anoxic waters of the Framvaren Fjord, Norway, as a starting point. The water column in this fjord is anoxic below 20 m depth, and at depths greater than 100 m, it is characterised by $H_2S$ concentrations between about 4.5 and 5.8 mM, as well as $NH_4^+$ concentrations between about 1.3 and 1.6 mM. Yao and Millero (1995) provide data for all the acid-systems currently considered in the Fortran 90 implementation of SOLVESAPHE-r2.

For the new test case ABW5, where "ABW" stands for *anoxic brackish water*, I then use average concentrations between 100 and 170 m depth for all components except $Alk_T$ and $C_T$, for which roughly rounded ranges over that depth interval are adopted: $T = 7.56\,°C$, $S = 22.82$, $D = 135\,m$, $[H_2S] = 5.1\,mmol\,kg^{-1}$, $[PO_4^{3-}] = 0.1\,mmol\,kg^{-1}$, $[NH_4^+] = 1.5\,mmol\,kg^{-1}$, $[SiO_2] = 0.6\,mmol\,kg^{-1}$, $Alk_T = 17 - 20\,mmol\,kg^{-1}$, $C_T = 15 - 17.5\,mmol\,kg^{-1}$. All reported concentrations are assumed to represent total concentrations of their respective acid systems.

In order not to lengthen the manuscript unnecessarily, the results for SW1 will be removed from Figs. 5 and 6 and those for ABW5 included instead. SW1 is a subset of SW2 and their iteration number histograms are broadly similar in terms of frequencies (not absolute numbers, as SW1 is based upon a $C_T$-$Alk_T$ grid with $600 \times 600$ points and SW2 upon one with $1500 \times 1300$ points). The results for SW1 are still going to be reported in the "*Additional Results*" in the Supplement.

**Minor Points and Technical Comments**

All the typos will be corrected as suggested and I am not mentioning them here.

**Anonymous reviewer # 1**

*Units of alkalinity: I suggest replacing meq (outdated) by mmol (compare, for example, Dickson et al., 2007, Chapter 5, Table 2)*

OK, will be amended.

*L51 's is a factor to convert from that scale to the free scale': it might be useful to mention that the value of $s$ is close to 1*

The sentence at lines 51–52 will be rewritten to read

"$s$ depends on temperature, pressure and salinity of the sample and its value is close to 1 (typically between 1.0 and 1.3)."

[Figure]

Figure R2: (a) $p$H isolines; (b) $CO_3^{2-}$ concentration isolines in $C_T$-$Alk_T$ space; (c) $CO_2$, (d) $HCO_3^-$ and (e) $CO_3^{2-}$ and concentration isolines in $p$H-$Alk_T$ space. These distributions were calculated with SOLVESAPHE version 1.0.3. For (c), (d) and (e), carbonate alkalinity, $Alk_C$, was derived by using eq. (2) *[from the manuscript]*, combined with eqs. (7), (8) and (9) *[from the manuscript]* to derive $[CO_2]$ $[HCO_3^-]$, and $[CO_3^-]$, respectively. Blank areas represent the $p$H-$Alk_T$ combinations that lead to negative $Alk_C$. Figure 3 in Deffeyes (1965) is similar to (b).

*Fig. 1: axes labels (quantities & units) missing, same for color bars; remove titles (numbers); y-axes from 1 to 0 to 3 or from −1 to 3 (???); a bit more explanation/discussion might be in order*

The annotations of Fig. 1 in the manuscript were partly lost during the processing of the submitted manuscript file (where the figure was complete) to produce the preprint posted on the GMDD forum. An Author's Comment (AC1) with a reprocessed version of the figure was posted on 9th March 2021 (doi:10.5194/gmd-2020-447-AC1). I reproduce the complete figure here as Fig. R2, in the version that I plan to include in the revised manuscript. It has its colour scheme changed for a colour-blind safe one and the panels in the lower row of the figure have been rearranged so that they are in $CO_2 - HCO_3^- - CO_3^{2-}$ order.

*L193 $[H^+] \gg$ : something missing here*

This was meant to be understood as "for great values of $[H^+]$". This will be replaced by "as $[H^+] \to +\infty$."

*L210 $H_1 < H_{min}$ and $H_2 > H_{min}$ might be shortened to $H_1 < H_{min} < H_2$*

Yes, this is correct and both forms of course mathematically equivalent. I prefer to leave it as is, to emphasize that $H_1$ is lower than $H_{min}$ and that $H_2$ is greater than $H_{min}$.

*L229 'exact knowledge determination' ???*

"determination" needs to be deleted so that the sentence reads "[...] for which the exact knowledge of $H_{tan}$ is not indispensable."

*L271 API = ???*

API is a standard acronym in computer science standing for "Application Programming Interface" – the definition will be added.

*L275 'In the course of the development s related to . . . ' ??? something missing here?*

There is actually a spurious blank between "development" and the "s" that follows. This should read "In the course of the developments related to [. . . ]"

*L291 'equation function' ???*

I make a distinction between an equation and the function that defines it, which I then call the equation function (e. g., when it comes to stating that the function defining the equation is monotonous or decreasing – see also line 179). Here "function" can nevertheless be discarded.

**Anonymous reviewer # 2**

*Throughout*

"on Fig. $n$" changed to "in Fig. $n$" as suggested repeatedly.

*L.10-11: "longer"/"more time"$--$> than what exactly?*

There is actually an error on line 11: "while $Alk_T$ & $CO_2$ requires about four times as much time." should actually read "while $Alk_T$ & $CO_3^{2-}$ requires about four times as much time." Lines 10–11 will be rewritten to read:

"The $Alk_T$ & $CO_2$ pair is numerically the most challenging. With the Newton-Raphson based solver, it takes about five times as long to solve as the companion $Alk_T$ & $C_T$ pair; the $Alk_T$ & $CO_3^{2-}$ pair requires on average about four times as much time as the $Alk_T$ & $C_T$ pair."

*L.12-13: "It outperforms the Newton-Raphson based one by a factor of four'$--$> In terms of what, calculation time?*

In terms of the required number of iterations. This will be reformulated more precisely.

*L.15: "For $Alk_T$ & $CO_3^{2-}$ data pairs" would read better here*

OK – will be corrected as suggested.

*L.27-29: Depending on the purpose, some modellers will use pH in combination with $C_T$; I suggest to write "most modellers" instead.*

OK – will be changed as suggested.

*L.38-39: Not sure what is meant with "this best had to be one pair of input data only".*

This means that users should only have to provide the absolutely necessary information (i. e., the pair of data), but no further auxiliary information, such as a bracketing interval or starting values for an iterative process. As in SOLVESAPHE v. 1, the algorithm should be able to derive that kind of information autonomously without having to rely on user input.

*L.40-44: I would suggest to finish the introduction and start a new manuscript section after presenting the aim.*

> The section heading "2 Theoretical Considerations" will be moved before the current line 44. The now initial part of that section will be headed by a new subsection title "2.1 Revisiting the mathematics of the alkalinity-$p$H equation" and rephrased for a smoother start.

*L.187: better write "I" instead of "we" (single author)*

> OK.

*L.193: [H + ] >> (something appears to be missing here)*

> See reply to the same comment by AR#1.

*L.270-271: I suggest to provide one sentence here to explain the difference between both solvers, for example by moving the current L.324-326 which explains that one is the Newton-Raphson solver, while the other uses the secant scheme.*

> The paragraph starting at line 270 will be reformulated along the following lines:

> "The SOLVESAPHE Fortran 90 library from Munhoven (2013) – hereafter SOLVESAPHE v. 1 – has been revised, cleaned up and upgraded to allow the processing of the additional three pairs. For the purpose of this paper, only the two main solvers have been kept: these are `solve_at_general`, which uses a Newton-Raphson method, and `solve_at_general_sec`, which uses the secant method. Both can be still be used with the same Application Programming Interface (API) as in v. 1. The instances in SOLVESAPHE-r2 are nevertheless only wrappers to the newly added Newton-Raphson based `solve_at_general2` and secant (or more precisely regula falsi) based `solve_at_general2_sec` both of which are able to process problems that have two roots. They return the number of roots of the problem, as well as their actual values, if any."

**References**

Deffeyes, K. S.: Carbonate Equilibria : A Graphic and Algebraic Approach, Limnol. Oceanogr., 10, 412–426, https://doi.org/10.4319/lo.1965.10.3.0412, 1965.

Dickson, A. G., Sabine, C. L., and Christian, J. R., eds.: Guide to Best Practices for Ocean $CO_2$ Measurements, vol. 3 of *PICES Special Publication*, Carbon Dioxide Information and Analysis Center, Oak Ridge (TN), URL `https://cdiac.ess-dive.lbl.gov/ftp/oceans/Handbook_2007/Guide_all_in_one.pdf`, 2007.

Munhoven, G.: Mathematics of the total alkalinity-pH equation – pathway to robust and universal solution algorithms: the SolveSAPHE package v1.0.1, Geosci. Model Dev., 6, 1367–1388, https://doi.org/10.5194/gmd-6-1367-2013, 2013.

Sharp, J. D. and Byrne, R. H.: Carbonate ion concentrations in seawater: Spectrophotometric determination at ambient temperatures and evaluation of propagated calculation uncertainties, Mar. Chem., 209, 70–80, https://doi.org/10.1016/j.marchem.2018.12.001, 2019.

Yao, W. and Millero, F. J.: The Chemistry Of the Anoxic Waters in the Framvaren Fjord, Norway, Aquat. Geochem., 1, 53–88, https://doi.org/10.1007/BF01025231, 1995.

Zeebe, R. E. and Wolf-Gladrow, D.: $CO_2$ in seawater : Equilibrium, kinetics, isotopes, vol. 65 of *Elsevier Oceanography Series*, Elsevier, Amsterdam (NL), URL `http://www.sciencedirect.com/science/bookseries/04229894/65`, 2001.

---

## Author Response (AR1)

**SolveSAPHE-r2: revisiting and extending the Solver Suite for Alkalinity-PH Equations for usage with $CO_2$, $HCO_3^-$ or $CO_3^{2-}$ input data**

**Author's Response**

Guy Munhoven

Guy.Munhoven@uliege.be

25th May 2021

Dear Sandra,

please find below my point to point listing of the changes made to the manuscript in response to the referees' comments and suggestions. For the sake of brevity, I do not repeat here the justifications that were given in the Author's Comments in reply to the Referees Comments. A *latexdiff* version of the manuscript highlighting the insertions and deletions in the text has also been uploaded alongside the revised manuscript.

I hope the manuscript is now acceptable for publication.

Best regards,

Guy

**Revisions in response to comments by Anonymous Referee #1**

**General Comments**

Anonymous Referee #1, hereafter AR#1, essentially has only one major comment, or more precisely question:

*When calculating pH from alkalinity and carbonate ion concentration in typical seawater [. . . ] one obtains two solutions and the question is 'Which one to take?' In order to shed some light on this problem I have calculated the concentrations of all carbonate system parameters for a fixed alkalinity ($Alk_T = 2300 \, \mu mol \, kg^{-1}$) as a function of pH (Fig. 1). For $[CO_3^{2-}] = 500 \, \mu mol \, kg^{-1}$ one obtains two solutions: $pH_1 = 8.70$ and $pH_2 = 10.75$ with quite different DIC values. The higher pH value is discarded because the DIC is 'unrealistically' low and that $pH = 8.70$ would be the chosen answer. This [choice] has been taken, for example, by Zeebe & Wolf-Gladrow (2001, p. 277, 'use the larger one'): they recommend using the real solution with the largest $H^+$ concentration (lowest pH). Although this choice works out fine in common seawater, it has not been properly justified. And the justification is actually beyond mathematics.*

Added Figure with a graph similar to the Figure provided by AR#1, but adapted to the results presented here (more complex $\text{Alk}_\text{T}$ composition, more extended $p\text{H}$ range.

In the manuscript I wrote that the existence of two roots for the $\text{Alk}_\text{T}$ & $\text{CO}_3^-$ pair system was a little known fact. As AR#1 illustrates "little known" does not mean "unknown"... Zeebe and Wolf-Gladrow (2001, pp. 276–277) did mention it: "Roots: two positive (use the larger one), three negative." I have added a mentioning of this in the manuscript (line [XXX] in the *latexdiff* report)

Added extra discussion regarding the choice of the root

**Specific/technical comments**

*Units of alkalinity: I suggest replacing meq (outdated) by mmol (compare, for example, Dickson et al., 2007, Chapter 5, Table 2)*

OK, corrected as recommended.

*L50 typo: whoe ? whose*

OK, corrected.

*L51 's is a factor to convert from that scale to the free scale': it might be useful to mention that the value of s is close to 1*

The sentence at lines 51–52 has been rewritten as follows:

"$s$ depends on temperature, pressure and salinity of the sample and its value is close to 1 (typically between 1.0 and 1.3)."

*Fig. 1: axes labels (quantities & units) missing, same for color bars; remove titles (numbers); y-axes from 1 to 0 to 3 or from $-1$ to 3 (???); a bit more explanation/discussion might be in order.*

The annotations of Fig. 1 in the manuscript were partly lost during the processing of the submitted manuscript file (where the figure was complete) to produce the preprint posted on the GMDD forum.

*L174 typo: eq. (12then $\rightarrow$ eq. (12) then*

OK, corrected.

*L193 $[\text{H}^+] \gg$ : something missing here*

This has been replaced by "as $[\text{H}^+] \rightarrow +\infty$."

*L210 $H_1 < H_\text{min}$ and $H_2 > H_\text{min}$ might be shortened to $H_1 < H_\text{min} < H_2$*

I prefer to leave it as is—nothing changed.

*L229 'exact knowledge determination' ???*

"determination" has been deleted so that the sentence now reads "[...] for which the exact knowledge of $H_\text{tan}$ is not indispensable."

*L271 API = ???*

    API is a standard acronym in computer science and stands for "Application Programming Interface" – the definition has been added (line 393 in the *latexdiff*

*L275 'In the course of the development s related to . . . ' ??? something missing here?*

    There was a spurious blank between "development" and the "s" that follows. Corrected to read "In the course of the developments related to [. . . ]"

*L291 'equation function' ???*

    Discarded "function"

*Fig. legend: 'two roots The' dot missing after roots*

    This was about the caption of Fig. 7 (now Fig. 8), and has been corrected.

**Sup. Mathematical and Technical Details**

*Typos:*
*2.3.1 $B(OH_3) \rightarrow B(OH)_3$*
*2.3.3 $H_3(PO)_4 \rightarrow H_3PO_4$*
*2.3.4 $H_4(SiO)_4 \rightarrow H_4SiO_4$*

    OK, corrected.

**Revisions in response to comments by Anonymous Referee #2**

**General comments**

*Understandably, the manuscript is not written for the broader scientific community working on the carbonate system because of its technical focus, and overall it is well written. However, I do think that some efforts can be made in making the paper more appealing to a wider audience. For example, more context can be given as to why it is important to include solutions for the $CO_2$ - $Alk_T$, $HCO_3^-$ - $Alk_T$ and especially $CO_3^{2-}$ - $Alk_T$ pairs. While $pCO_2$ has already been a commonly measured parameter for decades, $CO_3^{2-}$ can currently be regarded as the fifth parameter that can be measured to describe the carbonate system. Recent adoption of direct $CO_3^{2-}$ measurements by experimentalists (e.g. Easley et al., 2013; doi:10.1021/es303631g or Patsavas et al., 2015; doi:10.1016/j.marchem.2014.10.015) actually provide scientific ground for this manuscript and this is even strengthened given that $CO_3^{2-}$ was found to be best paired with $Alk_T$ (or $C_T$ ; Sharp and Byrne, 2018; doi:10.1016/j.marchem.2018.12.001).*

> The introduction was reorganised, partly rewritten and extended along the lines proposed by Anonymous Referee #2 (hereafter AR#2).

*Another comment I have along the same line is that I felt that a discussion was lacking on which pH value to take in the case that there are two solutions for the $CO_3^{2-}$ - $Alk_T$ pair. Later I noticed that Reviewer 1 has this exact comment and worked this out very nicely in their comment. I would therefore suggest Guy Munhoven to take this point into account and perhaps even create a figure similar to that by Reviewer 1 in the manuscript. Such a figure would also aid the less technical reader (as well as the reader who has difficulties in interpreting Deffeyes diagrams) in understanding the importance of this work. However, I also agree with Reviewer 1 that a justification of this choice is probably beyond the scope of this manuscript.*

> - A two-panel figure was added (new Fig. 2 in the revised manuscript). The top panel is similar to the figure presented by AR#1, but has been completed to make it compatible with the alkalinity composition used to produce Fig. 1 (i.e., including borate alkalinity). The target $[CO_3^{2-}]$ line intersecting the $CO_3^{2-}$ distribution function is not shown in order to avoid overloading the graph. The second panel shows a selection of carbonate ion distribution functions obtained for different $Alk_T$ values.
> - The graphs of the new figure are discussed in the text, and the possibilities of zero, one or two roots explored.
> - Discussion about which one of the roots to chose when there are two has been added in a new section 2.4.2 (two pages).

*Finally, it might be interesting to include some other case studies. Specifically, I was thinking about pore waters where the concentrations of various acid-base systems may be higher, especially the relative contributions of non-carbonate bases to $Alk_T$.*

> I had difficulties to secure sufficiently complete data sets for porewater chemistry to design a realistic representative test case as requested by AR#2. As the purpose was to cover samples where "[...] the concentrations of various acid-base systems may be higher, especially the relative contributions of non-carbonate bases to Alk" I finally resorted to using the data of Yao and Millero (1995) for the anoxic waters of the Framvaren Fjord, Norway, as a starting point. The water column in this fjord is anoxic below 20 m depth, and at depths greater than 100 m, it is characterised by $H_2S$ concentrations between about 4.5 and 5.8 mM, as well as $NH_4^+$ concentrations between about

1.3 and 1.6 mM. Yao and Millero (1995) provide data for all the acid-systems currently considered in the Fortran 90 implementation of SOLVESAPHE-r2.

For the new test case ABW5, where "ABW" stands for *anoxic brackish water*, I then use average concentrations between 100 and 170 m depth for all components except $Alk_T$ and $C_T$, for which roughly rounded ranges over that depth interval are adopted: $T = 7.56\,°C$, $S = 22.82$, $D = 135\,m$, $[H_2S] = 5.1\,mmol\,kg^{-1}$, $[PO_4^{3-}] = 0.1\,mmol\,kg^{-1}$, $[NH_4^+] = 1.5\,mmol\,kg^{-1}$, $[SiO_2] = 0.6\,mmol\,kg^{-1}$, $Alk_T = 17 - 20\,mmol\,kg^{-1}$, $C_T = 15 - 17.5\,mmol\,kg^{-1}$. All reported concentrations are assumed to represent total concentrations of their respective acid systems.

In order not to lengthen the manuscript unnecessarily, the results for SW1, which covers a subset of SW2) have been removed from Figs. 5 and 6 (now Figs. 6 and 7) and those for ABW5 included instead.

**Specific comments**

*Throughout*

"on Fig. $n$" changed to "in Fig. $n$" as suggested repeatedly.

*L.10-11: "longer"/"more time"$--$> than what exactly?*

There was actually an error on line 11: "while $Alk_T$ & $CO_2$ requires about four times as much time." should actually have read "while $Alk_T$ & $CO_3^{2-}$ requires about four times as much time."

Lines 10–11 have been rewritten to read (see lines 10–12 in the *latexdiff* report):

"The $Alk_T$ & $CO_2$ pair is numerically the most challenging. With the Newton-Raphson based solver, it takes about five times as long to solve as the companion $Alk_T$ & $C_T$ pair; the $Alk_T$ & $CO_3^{2-}$ pair requires on average about four times as much time as the $Alk_T$ & $C_T$ pair."

*L.12-13: "It outperforms the Newton-Raphson based one by a factor of four'$--$> In terms of what, calculation time?*

In terms of the required number of iterations. This has been reformulated more precisely as

"It outperforms the Newton-Raphson based one by up to a factor of four in terms of average numbers of iterations and execution time and yet reaches equation residuals that are up to seven orders of magnitude lower."

*L.15: "For $Alk_T$ & $CO_3^{2-}$ data pairs" would read better here*

OK – changed as suggested.

*L.27-29: Depending on the purpose, some modellers will use pH in combination with $C_T$; I suggest to write "most modellers" instead.*

OK – changed as suggested.

*L.38-39: Not sure what is meant with "this best had to be one pair of input data only".*

This means that users should only have to provide the absolutely necessary information (i. e., the pair of data), but no further auxiliary information, such as a bracketing interval or starting values for an iterative process. As in SOLVESAPHE v1, the algorithm should be able to derive that kind of information autonomously without having to rely on user input.

Lines 36–39 (see lines 60–67 in the *latexdiff* report) have been rewritten:

"Here, we do not focus on these aspects, but on the design of algorithms that can solve the underlying mathematical problem with as little user input as possible. The aim is to reduce user input to the bare essentials: besides the fundamental information about temperature, salinity, pressure and the thermodynamic data, this ideally had to be any physically meaningful data pair only; the algorithm should be able to derive any other auxiliary information, such as root brackets or starting values for iterations, on its own."

*L.40-44: I would suggest to finish the introduction and start a new manuscript section after presenting the aim.*

The section heading "2 Theoretical Considerations" has been moved before the earlier line 44, followed by a new subsection title "2.1 Revisiting the mathematics of the alkalinity-$p$H equation" The first sentence of the new subsection has been reformulated.

*L.50: "whose"*

OK – changed as suggested.

*L.187: better write "I" instead of "we" (single author)*

Actually on L. 188, but changed as suggested anyway.

*L.193: [H + ] >> (something appears to be missing here)*

Replaced by $[\mathrm{H}^+] \to +\infty$.

*L.270-271: I suggest to provide one sentence here to explain the difference between both solvers, for example by moving the current L.324-326 which explains that one is the Newton-Raphson solver, while the other uses the secant scheme.*

The paragraph starting at line 270 has been rewritten (see lines 389–397 in the *latexdiff* report):

"The SOLVESAPHE Fortran 90 library from Munhoven (2013) – hereafter SOLVESAPHE v. 1 – has been revised, cleaned up and upgraded to allow the processing of the additional three pairs. For the purpose of this paper, only the two main solvers have been kept: these are `solve_at_general`, which uses a Newton-Raphson method, and `solve_at_general_sec`, which uses the secant method. Both can be still be used with the same Application Programming Interface (API) as in v1. The instances in SOLVESAPHE-r2 are nevertheless only wrappers to the newly added Newton-Raphson based `solve_at_general2` and secant (or more precisely regula falsi) based `solve_at_general2_sec` both of which are able to process problems that have two roots. They return the number of roots of the problem, as well as their actual values, if any."

*L.275 "developments"*

OK – corrected.

*L.290: $CO_3^{2-}$ instead of $CO_3^{-2}$*

OK – corrected.

**Author's own changes**

**Revised graphs**

Jean-Pierre Gattuso has drawn my attention to issues with the rainbow colour scheme that I had adopted for most of my figures. The colour schemes of all the figures in the manuscript and in the Supplement have been changed from the rainbow scheme to a colour-blind friendlier one.

**Test cases: naming scheme and inter-comparison set-up**

*There was an inconsistency in the naming of the test cases: SW4 actually is for brackish water and not seawater, as the "SW" in the name might suggests.*

SW4 has been renamed to BW4, where "BW" stands for brackish water.

*While setting up the new test case ABW5 I realised that the currently defined ones are not consistent when it comes to comparing the computational requirements of the $Alk_T$ & $C_T$, $Alk_T$ & $CO_2$, $Alk_T$ & $HCO_3^-$ and $Alk_T$ & $CO_3^{2-}$ versions against each other, for a each of SW1, SW2, SW3 and BW4. Although the $[CO_2]$, $[HCO_3^-]$ and $[CO_3^{2-}]$ ranges for each test case had been defined on the basis of their respective distributions calculated from the $Alk_T$-$C_T$ results, they did not cover exactly the same "samples." To make the results for the different pairs actually comparable, the test case definitions were therefore adapted for the inter-comparison of the performances of the four data pairs. Each test case is first carried out with the $Alk_T$-$C_T$ pair, for each set of temperature, salinity and pressure, and the results stored. For the other three pairs, the $p$H distribution obtained with the $Alk_T$-$C_T$ pair for the chosen set of temperature, salinity and pressure is first read in and the corresponding $[CO_2]$, $[HCO_3^-]$ or $[CO_3^{2-}]$ distributions calculated on the underlying $Alk_T$-$C_T$ grid. The so-obtained arrays of species concentrations are then used to define the set of $Alk_T$-$CO_2$, $Alk_T$-$HCO_3^-$ and $Alk_T$-$CO_3^{2-}$ data pairs the benchmark calculations. This way the experiments for the four different characteristic carbonate system concentrations cover exactly the same set of samples.*

The description of the test cases in Sect. 3.2.1 and the discussion in Sect. 3.2.2 have been adapted accordingly.

*Figure 1*

- The $HCO_3^-$ concentration in $C_T$-$Alk_T$ space was added between the $p$H and $[CO_2]$ panels.
- The panels in the lower row of Fig. 1 have been rearranged so that they are in $CO_2 - HCO_3^- - CO_3^{2-}$ order.
- Star symbols have been added to the two panels with the $CO_3^{2-}$ concentration distributions to illustrate a case with two roots (to extend the amendments suggested by the referees).
- The figure caption has been amended.

*Discussion of Figure 1 in the text*

Similarly to $Alk_T$ & $CO_3^{2-}$, the $C_T$ & $HCO_3^-$ pair generally has two $p$H roots, but the diagnosis is much more straightforward. This is now also discussed in the text and the solution recipe in Appendix A has been amended.

*Figures 7 and 8 (previously Figures 6 and 7)*

- Removed panel for test case SW1 (subset of SW2).
- Added panel for the new test case ABW5.
- Amended the maximum numbers of iterations to the slightly changed results as a result of the modified intercomparison setup

*Figure 8 (previously Figure 7)*

- Added an insert with a histogram to the figure to make better usage of the free blank space and to make the information quantitatively more expressive.
- Amended the caption to explain the insert.

*Appendix A: The direct cases*

The recipe for solving the $C_T$ & $HCO_3^-$ was not correct. Unlike for $C_T$ & $CO_2$ and $C_T$ & $CO_3^-$, the quadratic equation to solve for $[H^+]$ does not always have roots, and most often have two of them. The $HCO_3^-$ fraction in DIC does have to fulfil an additional constraint; being lower than 1 is not sufficient.

The recipe for $C_T$ & $HCO_3^-$ is now treated separately and presented and discussed in more detail.

*Supplement: Additional Results*

- Added Tables S2 and S3 with additional details about the test case definitions.
- Introduced ABW5 results into Tables S2 and S3 (now Tables S4 and S5).
- Added results for the new test case ABW5 where suitable: added new Figs. S5 and S12; added one panel to Figs. S10–S19 (now Figs. S13–S22).
- Added new Fig. S9 with results for the test case SW3-sc that had been missing.
- Fig. S20 (now Fig. S23): added histogram inserts to the four panels of the figure to make better usage of the free blank space and to make the information quantitatively more expressive and amended the caption to explain the insert.

**Minor changes**

*Title*

Added new code version number (v2.0.1) in the title to reflect code changes during the revision.

Throughout the manuscript and the Supplement:

- The manuscript text and the figure annotations has been revised to adhere more closely to the Copernicus house style (abbreviations such as "Fig.", Eq.", "Table" never being abbreviated, spelling of sulfur-related components and processes as sulf[*xyz*], power notation for units, etc.)

- Remaining "DIC" instances have been replaced by "$C_T$", which is the notation used everywhere else in the manuscript.

- Corrected a few English errors
- Added details of the medians and most probable numbers of iterations in section 3.2.2 (see lines 503–505 in the *latexdiff* report).

*Tables*

The table captions have been moved above the tables.

*Figure 4 (now Figure 5)*

The figure caption has been slightly reformulated.

*Code availability* section:

- Archived the codes on Zenodo and amended the section text accordingly, giving the references of the archives.
- Added notice that SOLVESAPHE-r2 has been ported to R and made available on the Comprehensive R Archive NEtwork (CRAN) under https://cran.r-project.org/package=SolveSAPHE

**References**

Munhoven, G.: Mathematics of the total alkalinity-pH equation – pathway to robust and universal solution algorithms: the SolveSAPHE package v1.0.1, Geosci. Model Dev., 6, 1367–1388, https://doi.org/10.5194/gmd-6-1367-2013, 2013.

Yao, W. and Millero, F. J.: The Chemistry Of the Anoxic Waters in the Framvaren Fjord, Norway, Aquat. Geochem., 1, 53–88, https://doi.org/10.1007/BF01025231, 1995.

Zeebe, R. E. and Wolf-Gladrow, D.: $CO_2$ in seawater : Equilibrium, kinetics, isotopes, vol. 65 of *Elsevier Oceanography Series*, Elsevier, Amsterdam (NL), URL `http://www.sciencedirect.com/science/bookseries/04229894/65`, 2001.